# Speed and segmentation control mechanisms characterized in rhythmically-active circuits created from spinal neurons produced from genetically-tagged embryonic stem cells

Matthew J Sternfeld[1,2], Christopher A Hinckley[1], Niall J Moore[1], Matthew T Pankratz[1], Kathryn L Hilde[1,3], Shawn P Driscoll[1], Marito Hayashi[1,2], Neal D Amin[1,3,4], Dario Bonanomi[1†], Wesley D Gifford[1,4,5], Kamal Sharma[6], Martyn Goulding[7], Samuel L Pfaff[1*]

[1]Gene Expression Laboratory, Howard Hughes Medical Institute, Salk Institute for Biological Studies, La Jolla, United States; [2]Biological Sciences Graduate Program, University of California, San Diego, La Jolla, United States; [3]Biomedical Sciences Graduate Program, University of California, San Diego, La Jolla, United States; [4]Medical Scientist Training Program, University of California, San Diego, La Jolla, United States; [5]Neurosciences Graduate Program, University of California, San Diego, La Jolla, United States; [6]Department of Anatomy and Cell Biology, University of Illinois at Chicago, Chicago, United States; [7]Molecular Neurobiology Laboratory, Salk Institute for Biological Studies, La Jolla, United States

*For correspondence: pfaff@salk.edu

Present address: †Division of Neuroscience, San Raffaele Scientific Institute, Milan, Italy

Competing interests: The authors declare that no competing interests exist.

**Abstract** Flexible neural networks, such as the interconnected spinal neurons that control distinct motor actions, can switch their activity to produce different behaviors. Both excitatory (E) and inhibitory (I) spinal neurons are necessary for motor behavior, but the influence of recruiting different ratios of E-to-I cells remains unclear. We constructed synthetic microphysical neural networks, called circuitoids, using precise combinations of spinal neuron subtypes derived from mouse stem cells. Circuitoids of purified excitatory interneurons were sufficient to generate oscillatory bursts with properties similar to in vivo central pattern generators. Inhibitory V1 neurons provided dual layers of regulation within excitatory rhythmogenic networks - they increased the rhythmic burst frequency of excitatory V3 neurons, and segmented excitatory motor neuron activity into sub-networks. Accordingly, the speed and pattern of spinal circuits that underlie complex motor behaviors may be regulated by quantitatively gating the intra-network cellular activity ratio of E-to-I neurons.

## Introduction

Many behaviors are based on circuits with flexible activity capable of switching their output (*Bargmann and Marder, 2013*; *Garcia-Campmany et al., 2010*). Although connectomes and functional roles for the neuronal subtypes that comprise circuits have begun to be defined, the output of large multicellular networks are difficult to predict from the input pattern because the mechanisms that coordinate and regulate these complex systems remain poorly understood. A remarkable attribute of many CNS networks with extensive interconnections among the cells is their ability to default

**eLife digest** The nerve cells or neurons within an animal's nervous system connect with one another like the wires in a complex circuit. Each neuron can send and receive signals and a major challenge in neuroscience is to understand how these circuits of neurons behave. To do this, researchers often use genetic tools and computer modeling to map the connections between the cells in a nervous system. However, it remains difficult to predict how an input signal will appear at the output after it passes through a network made of different types of neuron.

Brains contain many networks of interconnected neurons. Some of these networks send signals with a rhythmic pattern and typically drive repetitive movements such as breathing and walking. The networks are called central pattern generators (or CPGs for short). They contain both excitatory and inhibitory neurons and can generate rhythmic activity without any additional input. Nevertheless CPGs are not rigid, but can flexibly control when and how fast the muscles are activated to suit the animal's needs. It is thought the circuits are flexible because of the way excitatory and inhibitory neurons interact, but it is not known how these interactions define the behavior of the circuit.

Sternfeld et al. have now developed a new method to examine how the neurons that make up a circuit influence its activity. First, embryonic stem cells from mice were coaxed to develop into a number of subtypes of both excitatory and inhibitory neurons in the laboratory. These neurons were used to grow networks of neurons in a dish, named "circuitoids". The precise combination of subtypes of neuron was deliberately varied between each circuitoid, and Sternfeld et al. then studied how the different circuitoids behaved.

Several subtypes of excitatory neurons showed rhythmic bursts of activity, just like simple CPGs. Moreover, the ratio of excitatory to inhibitory neurons in the circuitoids was critical for establishing how fast and synchronized the bursts of activity were across the network. It is possible that the brain also uses this simple strategy of varying the ratio of excitatory to inhibitory neurons in circuits of neurons to generate complex, yet highly flexible, circuits with rhythmic activity. Further work will be needed to test this idea.

Finally, other researchers will hopefully be able to use this new approach to construct circuitoids and learn more about how the brain generates and controls rhythmic activity. It might also be possible to one-day transplant similar circuitoids into people to repair injured or diseased parts of a nervous system, or use circuitoids that resemble specific neurological disorders to screen for new treatments.

into a sustained self-organized oscillatory activity (*Buzsaki, 2006*). If only excitatory (E) neurons comprised these networks it is thought that inputs would trigger an avalanche of epileptic activity (*Buzsaki, 2006*). Thus, inhibitory (I) cells are considered important components of dynamic neural networks, because they can impose a regulated pattern on the system (*Brown, 1911*; *Buzsaki, 2006*; *Goulding et al., 2014*; *Grillner and Jessell, 2009*; *Isaacson and Scanziani, 2011*; *Marder and Bucher, 2001*).

Within neuronal networks, inhibitory neurons can increase coding efficiency, sharpen contrasts, and specify the topography of active circuits (*Arevian et al., 2008*; *Buetfering et al., 2014*; *Denève and Machens, 2016*). While the physiological roles of rhythmic brain activity are not always apparent, oscillatory central pattern generator (CPG) networks underlie breathing, chewing, scratching, and locomotion (*Grillner and Jessell, 2009*; *Marder and Bucher, 2001*). Vertebrate CPGs associated with locomotion are distributed networks of interconnected excitatory and inhibitory neurons that represent autonomous units capable of generating precisely patterned rhythmic activity (*Feldman and Smith, 1989*; *Nishimaru et al., 2000*; *Whelan et al., 2000*; *Cowley and Schmidt, 1995*; *Grillner, 2006*). Importantly, spinal CPGs have dynamic features that endow them with the ability to switch their frequency of rhythmicity in order to modulate locomotor speed and change the inter-coordination of motor pool firing patterns to drive different motor behaviors. The basis for producing flexible CPG activity remains unclear (*Grillner, 2006*; *McCrea and Rybak, 2008*), but it is likely founded in the differential recruitment of excitatory and inhibitory neurons comprising the CPG via selective inputs from descending and sensory systems (*Cowley and Schmidt, 1995*;

*Gosgnach et al., 2006*; *Grillner, 2006*; *Zhang et al., 2014*; *Zhong et al., 2011*). However, the contribution of a precisely controlled E/I cell ratio in the context of rhythmic circuit dynamics has been difficult to establish using cell ablation approaches that eliminate entire cell populations or pharmacological applications that silence entire types of synaptic transmission.

The molecular and functional characterization of spinal interneuron subtypes that contribute to vertebrate CPG activity has led to the identification of cells with local and long-range connections, with ipsilateral and contralateral projections, and with inhibitory and excitatory properties (*Alaynick et al., 2011*; *Bikoff et al., 2016*; *Gabitto et al., 2016*; *Garcia-Campmany et al., 2010*; *Goulding and Pfaff, 2005*; *Grillner and Jessell, 2009*; *Stepien and Arber, 2008*). A common trait among each class of spinal interneurons comprising the locomotor CPG is that they form an extensive network of interconnections within their subclass, between the interneuron subclasses, and onto motor neurons (*Alvarez et al., 2005*; *Crone et al., 2008*; *Lanuza et al., 2004*; *Zhang et al., 2008*). In addition to their connectivity, the functional role of interneuron subclasses has been investigated by genetic methods such as cell killing or silencing. Ablation of inhibitory V1 neurons slows the burst frequency of the CPG (*Gosgnach et al., 2006*), whereas ablation of both V1 and V2b inhibitory neurons disrupts the coordination of flexor-extensor alternation (*Zhang et al., 2014*). Removing the V2a or V3 excitatory neurons from the network results in more irregular CPG bursting (*Crone et al., 2008*; *Zhang et al., 2008*). These studies elegantly demonstrate that V1, V2b, V2a, and V3 interneurons each represent a necessary component of the CPG, but they do not address whether the cell types play instructive roles in gating the dynamic features of the circuitry. To begin to address this difficult issue, we used genetically labeled neurons to construct synthetic rhythmically active neuronal networks composed of defined cell subtypes. This approach allowed us to investigate whether quantitative differences in the cellular E/I ratio of specific neuronal subtypes are sufficient to alter the oscillatory speed and intra-network coordination of bursts.

Embryonic stem cells with genetic reporters for individual spinal cord interneuron subtypes and motor neurons were isolated and differentiated with inductive factors that mimic embryonic development of the spinal cord (*Kutejova et al., 2016*; *Peljto et al., 2010*; *Wichterle and Peljto, 2008*; *Wichterle et al., 2002*). We found that synthetic networks comprised of only excitatory V2a or V3 interneurons were sufficient to produce stable rhythmic bursts of activity at a frequency similar to drug-evoked fictive locomotor CPG activity. In contrast, V1 inhibitory neurons lacked the ability to produce regular bursts in isolation, but they did affect the activity of other cells. The addition of increasing numbers of V1 (I) neurons to fixed V3 (E) networks quantitatively accelerated the burst frequency, demonstrating that the E/I cell ratio instructively sets the oscillation speed. Interestingly, we found that V1 (I) neurons influenced the activity of motor (E) neuron bursting differently. Motor neuron networks, whose interconnected cells fire in unison, were uncoupled into sub-networks by the addition of increasing numbers of V1 cells. The formation of subnetworks is a form of patterning we define as segmentation. Taken together, our data lead to a model in which input-specific shifts in E/I cellular activity can flexibly tune the speed and pattern of an autonomous oscillatory circuit. By extension, this simple strategy for controlling the oscillatory properties that naturally emerge from interconnected neuronal networks might serve as the basis for generating complex, yet highly flexible, motor behaviors.

## Results

### De novo generation of spinal cord neurons

We constructed circuits with neurons that were generated de novo from mouse embryonic stem (ES) cells rather than using neurons isolated from spinal cords that may have acquired functional properties through complex interactions with their environment. Stem cell lines were isolated from blastula embryos harboring fluorescent reporters that indelibly label cardinal populations of spinal neurons as they develop postmitotically and form mature circuits. This allowed us to accurately and sensitively monitor ES cell differentiation into spinal neuron subtypes in real time. We isolated four independent ES cell lines containing the V1 reporter En1:Cre/tdTomato, two lines with the V2a reporter Chx10:Cre/tdTomato, seven lines with the V3 reporter Sim1:Cre/tdTomato, and two lines with the motor neuron reporter tgHb9-GFP (*Figure 1A*). These ES cell lines were differentiated into neurospheres containing ~50,000 aggregated cells using retinoic acid (RA) and smoothened agonist (SAG)

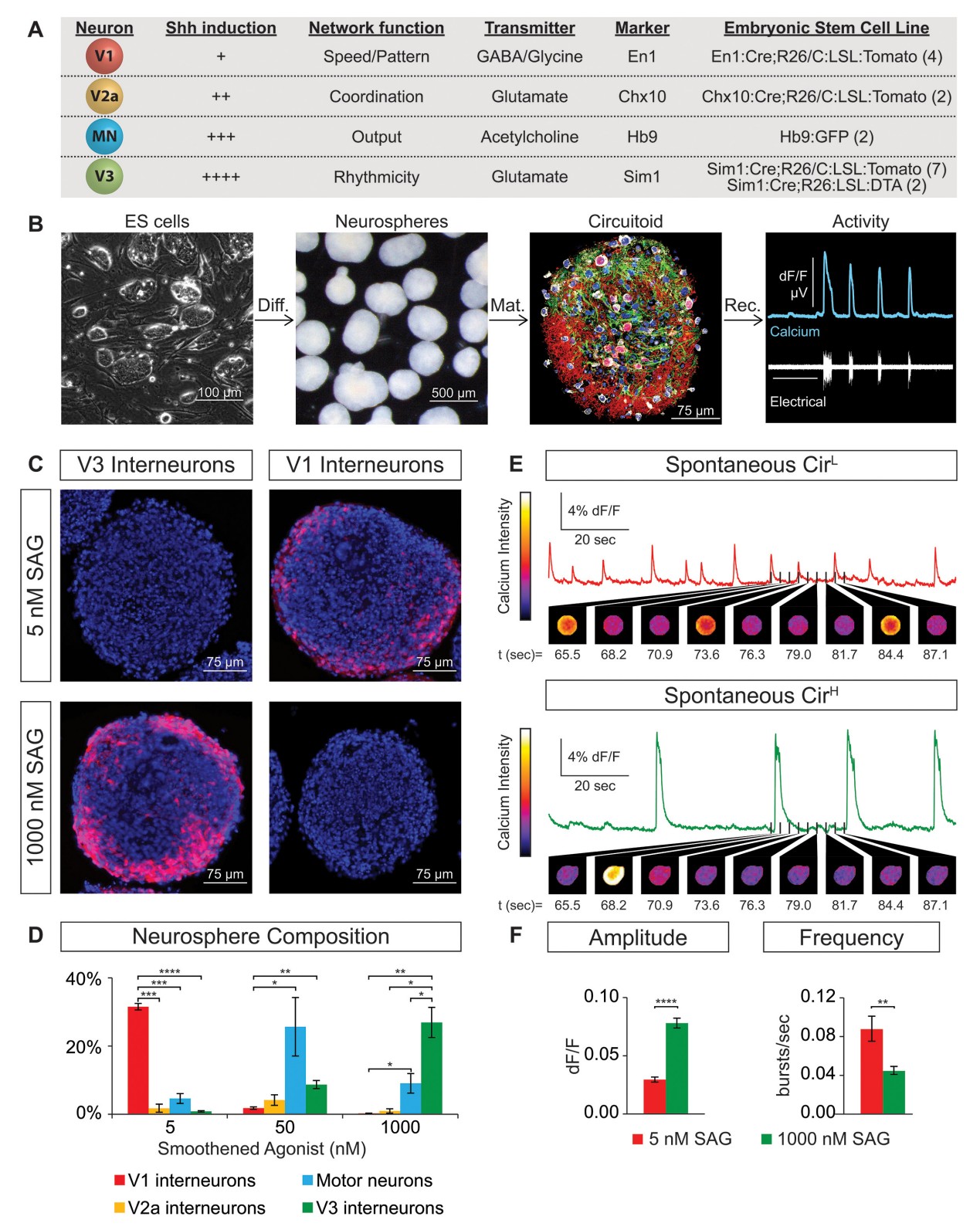

**Figure 1.** Spontaneous activity emerges from networks created from ES cell-derived spinal neurons. (**A**) Mouse ES cell lines were derived from embryos with genetic tags for defined spinal neuron subclasses. The number of individual ES cell lines generated for each genotype is shown in parentheses. (**B**) ES cells (phase contrast) were differentiated (diff.) into neurospheres (dark field) with retinoic acid and smoothened agonist (SAG) for six days. After maturation (mat., 17 days post-ES cell), a circuitoid was immunostained to identify cell types (neurotrace: white; V3 tomato reporter: red; astrocytes,

*Figure 1 continued on next page*

*Figure 1 continued*

GFAP: green; nuclei, DAPI: blue). Network activity was recorded (rec.) from circuitoids using a suction electrode and calcium imaging (tgCAG:GCaMP3 ES cell line, 1000 nM SAG, 15 days post-ES). Vertical scale bar is 8% dF/F and 280 μV, and horizontal 1 min. (C) The V3 reporter line generates tomato +V3 interneurons (red) with 1000 nM SAG, whereas the V1 ES cell line generates tomato +V1 interneurons (red) with 5 nM SAG 10 days post-ES cell differentiation. DAPI (blue). (D) Quantification of neuronal subtypes from ES cell-fluorescent reporter lines in (A) generated with increasing SAG concentrations using FACS to quantify cell numbers. Mean ± standard error of the mean (SEM). Differentiations: n = 8 for each ES cell-reporter line tested at each SAG concentration. Unpaired t test: *p<0.05; **p<0.01; ***p<0.001; ****p<0.0001. (E and F) Circuitoids composed of different neuronal subtypes produce distinct spontaneous network burst activity. (E) Calcium dye traces from spontaneously active circuitoids (Cir) generated at low (L) or high (H) SAG concentration: 5 nM SAG, Cir$^L$, red trace; 1000 nM SAG, Cir$^H$, green trace. Fluorescent images shown for indicated time points. (F) Quantification of bursting parameters using calcium imaging. Cir$^L$ spheres produce low-amplitude high-frequency bursts compared to Cir$^H$ spheres 16–17 days post-ES. Mean ± SEM; 1000 nM n = 21; 5 nM n = 19; unpaired t test ***p<0.001, ****p<0.0001.

The following figure supplements are available for figure 1:

**Figure supplement 1.** Neurosphere differentiation and composition.

**Figure supplement 2.** ES-cell-derived neurons express neuronal subtype markers.

**Figure supplement 3.** Circuitoid activity and synaptic structures.

following procedures similar to those described for generating motor neurons (*Peljto et al., 2010*; *Wichterle and Peljto, 2008*; *Wichterle et al., 2002*). These ES-cell-derived neurospheres could be maintained for weeks in culture and were found to contain glia and multiple neuronal subtypes (*Figure 1B*). These cellular aggregates did not appear to adopt a morphological organization that resembled the neural tube or spinal cord, rather neurons and glia seemed to be randomly distributed within the spheres (*Figure 1B*, data not shown).

ES cells with each neuronal-subtype reporter were differentiated with increasing SAG concentrations and the resulting neurospheres were dissociated and quantified using FACS (fluorescent activated cell sorting)(*Figure 1—figure supplement 1A,B*). At 5nM SAG, dorsally located V1 interneurons were enriched compared to ventral V3 interneurons and motor neurons (*Figure 1C and D*). V3 interneurons were preferentially generated at the expense of V1 interneurons in 1000 nM SAG, whereas motor neurons were most enriched using 50–200 nM SAG (*Figure 1C and D*). In each case, up to 30% of the cells within a neurosphere were V1, V3, or motor neurons using the optimal SAG concentration for each neuronal subtype, with similar results observed for each of the independent stem cell lines. V2a interneurons proved more difficult to generate and variability among our two reporter lines for these cells was noted (*Figure 1—figure supplement 1C*). We found, however, that blocking the notch-delta signaling pathway with the gamma secretase inhibitor DAPT improved V2a interneuron generation consistent with previous developmental studies on specification of this interneuron subtype (*Del Barrio et al., 2007*; *Brown et al., 2014*; *Crone et al., 2008*) (*Figure 1—figure supplement 1D*). In addition to the cardinal marker used to label each cell type (*Figure 1A*), RNA-sequencing of FACS purified ES-cell-derived motor neurons and V2a interneurons revealed that each expressed a battery of genes consistent with their identity assignment (*Al-Mosawie et al., 2007*; *Brown et al., 2014*; *Kimura et al., 2006*; *Kutejova et al., 2016*; *Lundfald et al., 2007*; *Saueressig et al., 1999*; *Wichterle et al., 2002*; *Xu and Sakiyama-Elbert, 2015*; *Zhang et al., 2008*) (*Figure 1—figure supplement 2*). ES-cell-derived V1 and V3 interneuron identity was monitored by assaying for expression of their cardinal marker gene (En1 and Sim1, respectively), and confirming their neurotransmitter identity (glycinergic/GABAergic and glutamatergic, respectively).

## Circuitoids produce coordinated bursts of activity

The neuronal subtypes within the ventral spinal cord form extensive interconnections with each other and are known to fire spontaneous bursts of activity (*Alvarez et al., 2005*; *Barry and O'Donovan, 1987*; *Crone et al., 2008*; *Lanuza et al., 2004*; *O'Donovan and Landmesser, 1987*; *Zhang et al., 2008*). We observed that ES-cell-derived neurons formed numerous excitatory synapses in culture (*Figure 1—figure supplement 3C*). To investigate whether matured neurospheres likewise produce spontaneous bursts, we monitored network output using both an extracellular suction recording

electrode and calcium imaging dyes and found the cell permeable calcium dyes provided an accurate measure of neuronal activity (*Figure 1B*, *Figure 1—figure supplement 3A and B*). Likewise, genetically encoded ubiquitously expressed, GCaMP3 was also a reliable reporter of activity (*Figure 1B*); however, most experiments were performed with cell-permeable calcium indicator dyes because they were easier to use. Imaging neuronal activity revealed that bursting within the spheres was highly synchronized across many neurons, suggesting the cells were interconnected (*Video 1*). We designated these de novo networks 'circuitoids' because they displayed a highly organized and behaviorally relevant pattern of activity. This nomenclature is intended to help distinguish these microphysical systems from neurospheres defined more broadly by the presence of neurons, or embryoids and organoids, which are defined by their morphological and cellular organization rather than their neuronal activity (*Lancaster et al., 2013*; *Sato et al., 2009*; *Stevens, 1960*).

Next, we examined whether the cellular composition of the circuitoids influenced the spontaneous activity by comparing bursts recorded from circuitoids produced with high SAG concentrations (1000 nM, $Cir^H$, cell composition: V3>MN>V1) versus low SAG (5 nM, $Cir^L$, cell composition: V1>MN>V3)(cell types quantified *Figure 1D*). We found that $Cir^L$ spheres produced bursts with lower amplitude and higher frequency than $Cir^H$ spheres (*Figure 1C–F*). Since $Cir^H$ produced reliable high-amplitude bursts, these circuitoids were used to examine the development of and physiological basis for this neuronal activity. $Cir^H$ spheres are enriched in V3 interneurons but are nonetheless heterogenous mixtures of ventral-spinal cord cell types. We found that these circuitoids were spontaneously active at two weeks and bursting increased in frequency at week 5 (*Figure 2A*). Acetylcholine receptor antagonists did not significantly alter the spontaneous bursting of circuitoids (*Figure 2B*), whereas the glutamatergic AMPA receptor antagonist, CNQX, blocked the activity suggesting that excitatory synaptic drive is necessary for coordinated bursts (*Figure 2C*). Next, we examined how these circuitoids respond to NMA and 5-HT, which trigger rhythmic activity in CPG circuits (evoked condition, see Materials and methods, *Kudo and Yamada, 1987*; *Smith and Feldman, 1987*; *Whelan et al., 2000*). Remarkably, these drugs activate a long-lasting highly regular (rhythmic) pattern of activity from circuitoids that is similar in frequency to fictive locomotor preparations (*Figure 2D*). Taken together, these results indicate that circuitoids acquire network properties that allow the interconnected cells to fire rhythmic bursts.

## Cellular composition in circuitoids influences rhythmic activity

Circuitoids generated from low or high SAG concentrations during differentiation are biased toward distinct sets of neuronal subtypes, which correlate with specific alterations in functional output (see *Figure 1*). We noted, however, that while most $Cir^H$ spheres produced regular activity in the evoked condition, a minority had irregular bursting (*Figure 3A and D*). This led us to consider whether subtle differences in circuitoid composition may cause variability in their activity characteristics. We subdivided $Cir^H$ spheres into two populations for further analysis: $Cir^{H-R}$ (rhythmic, interval coefficient of variation (I.C. V.)<0.2) and $Cir^{H-NR}$ (non-rhythmic, I.C.V.>0.2). When picrotoxin and strychnine, two inhibitory antagonists, were applied, $Cir^{H-NR}$ spheres burst in a more regular pattern (*Figure 3A–C*), suggesting their irregular bursting was caused by the presence of inhibitory neurons. Consistent with this, $Cir^{H-R}$ spheres remained rhythmic in inhibitory antagonists; however, we noted that their burst frequency slowed (*Figure 3D–F*, see below). Thus, under baseline conditions $Cir^{H-R}$ and $Cir^{H-NR}$ spheres may differ slightly in their level of inhibitory neuron activity, accounting for their different bursting characteristics.

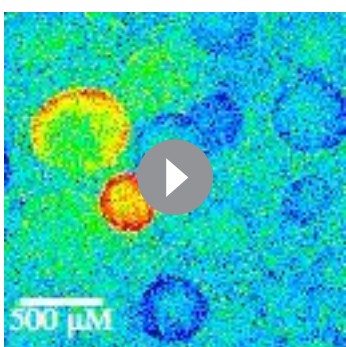

**Video 1.** Mature circuitoids display spontaneous activity. Heterogeneous circuitoids display spontaneous bursts of network activity that appear to be synchronous throughout each sphere. Here, an En1: Cre;R26/C:LSL:Tomato ES cell line was differentiated with 1000 nM SAG and allowed to mature until 16 days post-ES cell. Calcium intensity change (dF/F) was pseudocolored (scale from black to white). About 15 individual circuitoids with 50,000 cells each are in the field of view. Movie plays at 2x speed.

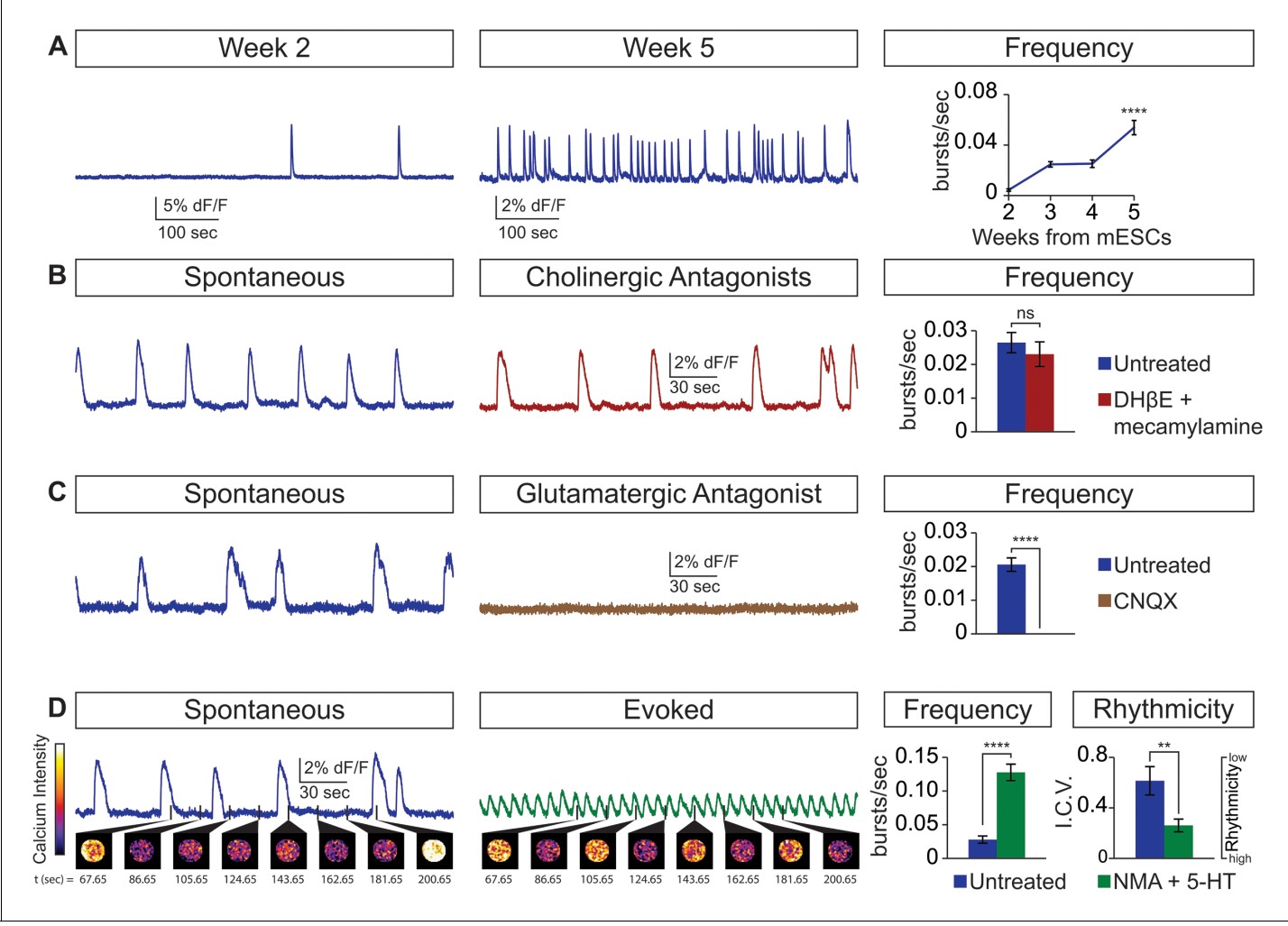

**Figure 2.** Physiological properties of circuitoids. Circuitoids were generated with 1000 nm SAG (Cir[H]) for analysis using calcium imaging with a ubiquitously expressed GCaMP3 ES cell line. (A) Spontaneous bursting frequency increased from week 2 to 5. Mean ± SEM, Week 2 n = 78; Week 3 n = 60; Week 4 n = 46; Week 5 n = 48; ****p<0.0001, unpaired t test. (B–D) Activity recorded 15–17 days post ES cell differentiation. (B) Cholinergic antagonists DHβE and mecamylamine (red) do not block activity. Mean ± SEM, n = 17 circuitoids, ns p=0.31, paired t test. (C) Glutamatergic antagonist CNQX (brown) abolishes bursting. Mean ± SEM, n = 16 circuitoids; ****p<0.0001, paired t test. (D) CPG evoking drugs NMA and 5-HT increase the frequency and rhythmicity of circuitoids compared to spontaneous activity. Rhythmicity measured using interval coefficient of variation (I.C.V.; low values indicate high rhythmicity). Mean ± SEM, n = 18 circuitoids; ****p<0.0001, **p<0.01, paired t test.

To test how circuitoid rhythmic bursting responds to a reduction in the cellular level of excitatory drive, we employed a genetic approach to ablate V3 excitatory neurons. Transgenic floxed-diphtheria toxin subunit-A (DTA) mice were crossed to Sim1-Cre animals and used to isolate ES cells in which V3 interneurons are killed upon differentiation (V3[DTA], *Figure 1A*) (*Zhang et al., 2008*). We found that V3[DTA] circuitoids failed to burst regularly in drugs that evoke rhythmic CPG activity compared to controls, and they produced these irregular bursts at a higher frequency (*Figure 3G–J*). Interestingly, by lowering the inhibitory drive in V3[DTA] circuitoids with inhibitory antagonists the networks switched to a slower and more regular activity pattern (*Figure 3K–N*). These findings reveal that the cellular composition within heterogeneous circuitoids influences the network's activity.

## Excitatory interneuron networks produce rhythmic activity

Despite extensive functional characterization of neuron subtypes within the spinal CPG, it remains unclear whether a particular cell type acts as a pacemaker, or if the rhythmicity arises from the

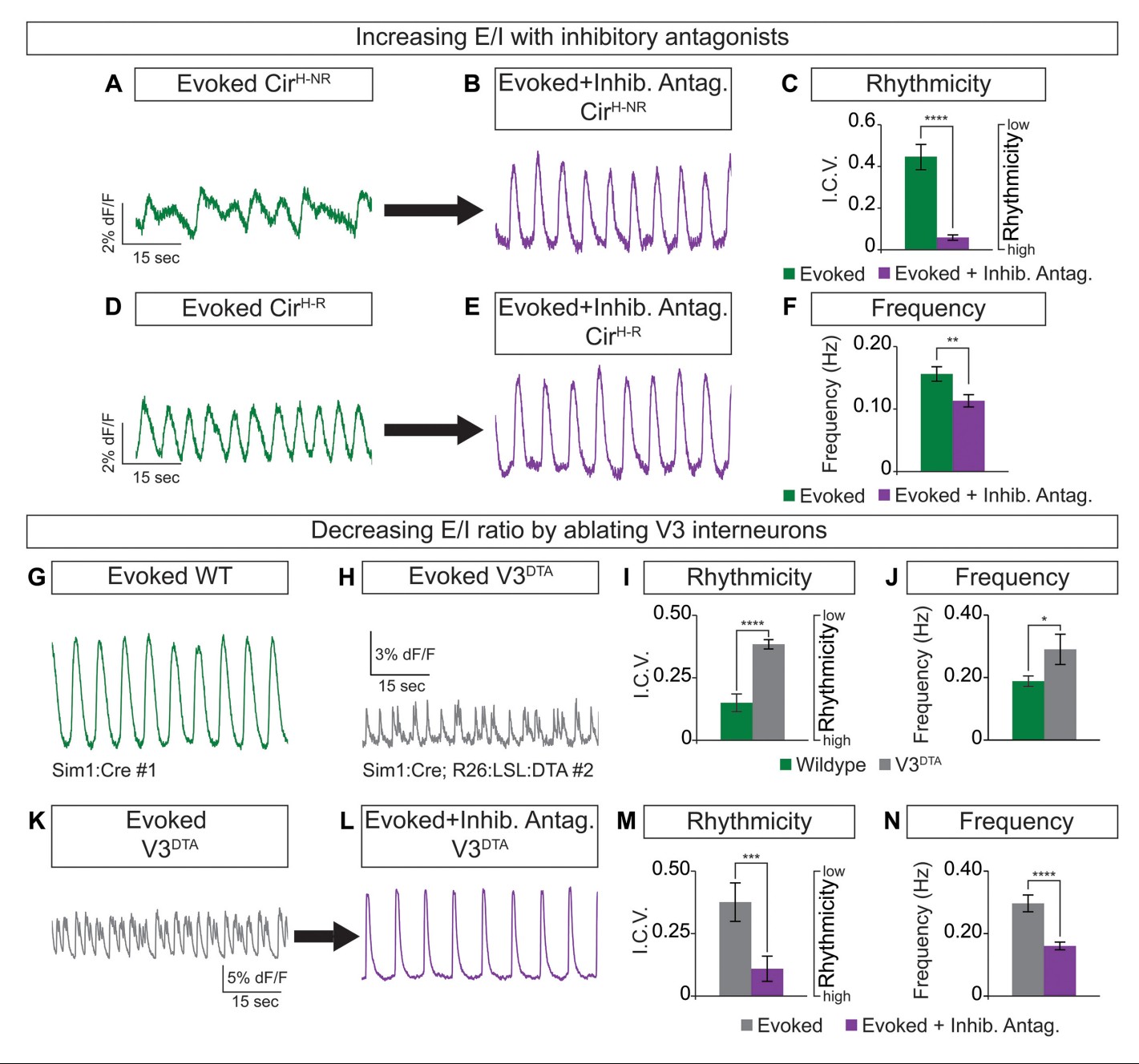

**Figure 3.** Cellular composition influences circuitoid rhythmicity and burst speed. (A–F) Circuitoids generated with 1000 nM SAG (Cir^H) were recorded in drugs that evoke CPG activity (evoked, NMA +5 HT, green) 16–17 days post ES cell differentiation. Activity was classified as either rhythmic (Cir^H-R, I.C. V.<0.2) or non-rhythmic (Cir^H-NR, I.C.V.>0.2). (A) Irregular bursting Cir^H-NR sphere. (B) Cir^H-NR bursting after application of inhibitory antagonists strychnine and picrotoxin (purple). (C) Cir^H-NR bursting becomes more rhythmic (lower I.C.V.) when inhibitory synaptic transmission is blocked. Mean ± SEM, n = 14 circuitoids, paired t test ****p<0.0001. (D) Bursting activity of Cir^H-R sphere. (E) Bursting activity of Cir^H-R after application of inhibitory antagonists strychnine and picrotoxin (purple). (F) Cir^H-R burst frequency decreases when inhibitory synaptic transmission is blocked. Mean ± SEM, n = 9 circuitoids, paired t test **p<0.01. (G–N) Circuitoids generated from the V3^DTA ES cell line (derived from Sim1:Cre;R26:LSL:DTA) with 1000 nM SAG (Cir^H) were recorded in drugs that evoke CPG activity (evoked, NMA +5 HT, green) 16–17 days post differentiation. (G) Control (Sim1:Cre, WT, green) circuitoid bursting. (H) V3^DTA bursting (grey) in circuitoids lacking V3 interneurons. (I) V3^DTA bursting is less rhythmic and (J) more frequent. (K–N) V3^DTA bursting becomes more rhythmic and less frequent with inhibitory antagonists strychnine and picrotoxin (purple). (G–J) WT n = 5 independent differentiations n = 69 circuitoids. V3^DTA n = 5 independent differentiations n = 85 circuitoids. Mean ± SEM, unpaired t test: *p<0.05, ****p<0.0001. (K–N) n = 18 circuitoids, mean ± SEM, paired t test: ***p<0.001; ****p<0.0001.

emergent properties of interconnected excitatory and inhibitory neurons (*Feldman et al., 2013*; *Harris-Warrick, 2010*; *Marder and Bucher, 2001*). We sought to determine whether synthetic networks generated from specific neuronal subtypes were sufficient to produce rhythmic activity.

Cir^H spheres were generated using the V3 interneuron reporter line, dissociated and tomato^+ V3 interneurons were purified with FACS (*Figure 4A*, *Figure 4—figure supplement 1A,B*). We reconstituted the purified V3 neurons into a synthetic network (purity > 98%) by aggregating these cells around an astrocyte core (*Figure 4A*, *Figure 4—figure supplement 1C*). Similar to heterogeneous circuitoids, large spontaneous bursts of activity were observed in synthetic networks comprised of only V3 neurons (*Figure 4B*). Likewise, NMA and 5-HT evoked a rhythmic pattern of activity with a frequency higher than the spontaneous burst rate (*Figure 4B, C, S, T*). Consistent with the glutamatergic properties of V3 interneurons, inhibitory antagonists did not significantly perturb the evoked pattern of bursting, whereas the glutamatergic antagonist, CNQX, blocked activity (*Figure 4C–E and R–T*). If gap junctions are present between cells, they appear to be insufficient to sustain rhythmic network activity in the presence of AMPA receptor blockers. As V3 circuitoids mature from 17 to 45 days after the initiation of ES cell differentiation, we determined that their bursting frequency increases (*Figure 4—figure supplement 2A*). We also found that circuitoids ranging in size from 5000 to 100,000 cells exhibited similar bursting parameters, suggesting they form highly scalable networks (*Figure 4—figure supplement 2B*). All together, these findings indicate that synthetic networks comprised of synaptically coupled, purified V3 interneurons are sufficient to produce rhythmic bursting in response to drugs that evoke CPG activity.

Like V3 interneurons, V2a cells represent another major class of excitatory glutamatergic interneurons found to contribute to robust CPG activity (*Crone et al., 2008*). We purified V2a interneurons (*Figure 4—figure supplement 1A,D*) and created synthetic networks that were spontaneously active and became rhythmic in the evoked condition (*Figure 4F, G and S*). The amplitude, rhythmicity, and frequency of V2a and V3 networks were similar (*Figure 4R–T*). As expected, the burst patterns of V2a excitatory networks were not affected by glycine/GABA antagonists but were disrupted by AMPA receptor blockers (*Figure 4G–I and R–T*).

Networks comprised of purified motor neurons (*Figure 4—figure supplement 1E*), which have recurrent collaterals that synapse onto neighboring motor neurons and interneurons (*Eccles et al., 1954*; *Nishimaru et al., 2005*; *Renshaw, 1946*), also displayed spontaneous bursting and responded to drugs that evoke CPG activity (*Figure 4J and K*). However, the bursts were more irregular and in general had greater variability between experiments than those produced by the V2a and V3 interneurons networks (*Figure 4S*). The activity in motor neuron circuitoids was dependent upon glutamatergic signaling rather than acetylcholine (*Figure 4K–M and T*, *Figure 4—figure supplement 3*). These findings are consistent with the intrinsic burst properties of motor neurons (*Hochman et al., 1994*; *Kiehn et al., 2000*; *MacLean et al., 1997*), their responsiveness to glutamatergic input (*Jahr and Yoshioka, 1986*; *Rekling et al., 2000*), and their co-release of cholinergic and glutamatergic neurotransmitters (*Herzog et al., 2004*; *Lamotte d'Incamps and Ascher, 2008*; *Meister et al., 1993*; *Mentis et al., 2005*; *Nishimaru et al., 2005*).

To test whether networks of inhibitory neurons can produce rhythmic activity, we generated circuitoids of V1 interneurons (*Figure 4—figure supplement 1A,F*). While spontaneous activity can be observed in V1 networks, the burst amplitudes are significantly lower than excitatory networks and fail to burst rhythmically in drugs that evoke CPG activity (*Figure 4N, O, R and S*). Although cell purifications are >98% pure (see *Figure 4—figure supplement 1C*), the low levels of activity found in V1 networks could be due to contamination with small amounts of excitatory neurons because inhibitory antagonists increased burst amplitude and rhythmicity, and a glutamatergic AMPA receptor blocker, CNQX, stopped the activity (*Figure 4N–Q and T*). These findings reveal a marked difference between the rhythmic activity of excitatory versus inhibitory neuron networks, and indicate that multiple subclasses of excitatory interneurons are sufficient to form rhythmically active networks.

## The cellular E/I ratio determines network burst speed

Dynamic circuits can flexibly switch how they use different cellular components within the larger network in order to meet the changing behavioral demands of the animal (*Ampatzis et al., 2014*; *McLean et al., 2007*, *2008*). This suggests that the active cellular E/I relationships within complex networks are not fixed, prompting us to explore how network activity is affected by different E/I cellular ratios. We differentiated neurons from ES cells and purified fluorescently tagged subtypes with

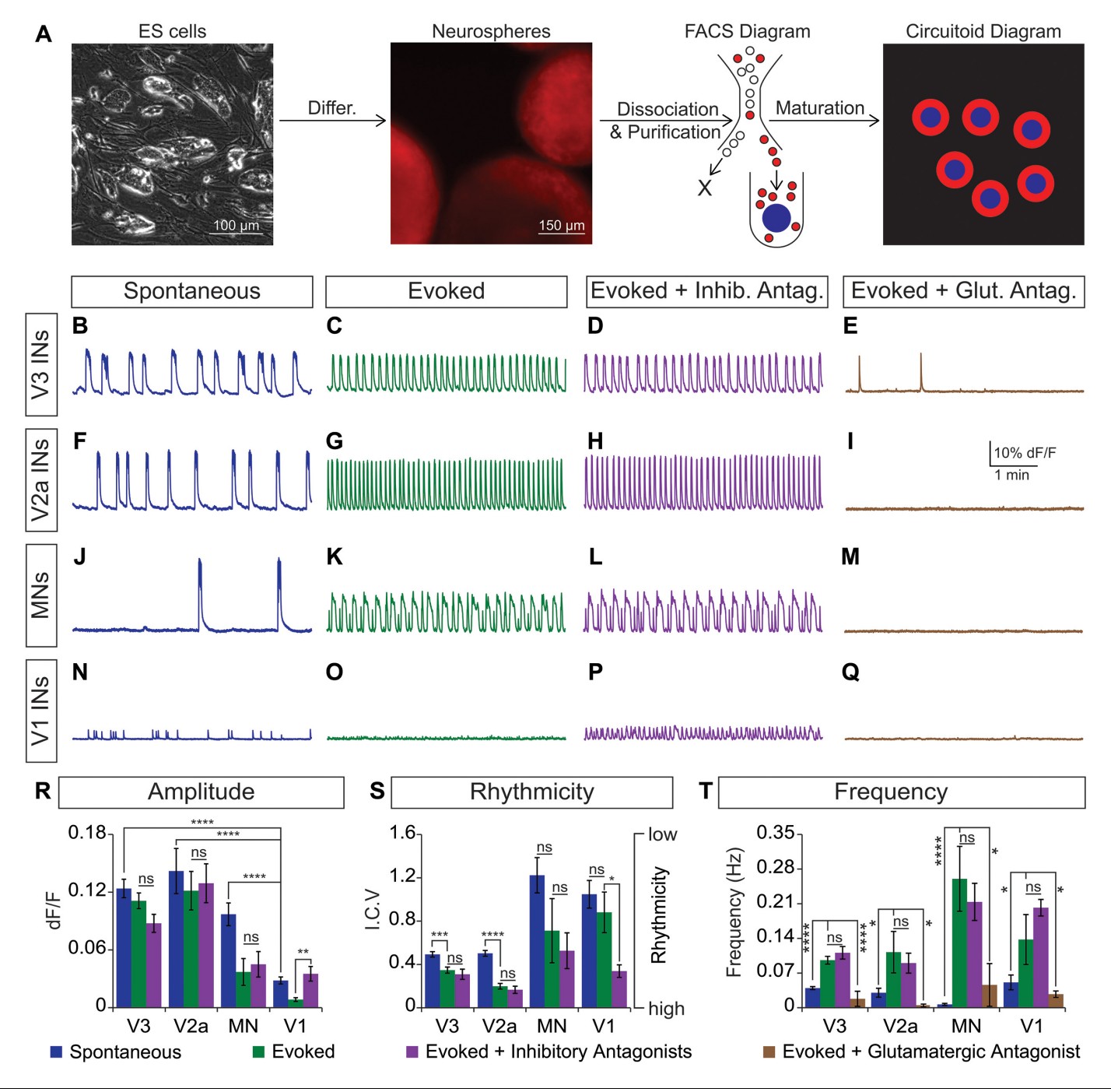

**Figure 4.** Rhythmic activity in networks with purified neuron subclasses. (**A**) Circuitoids of defined cellular composition were created by re-aggregating FACS purified ES cell-derived neurons (red) with astrocytes (blue) following the outlined scheme. (**B–Q**) Bursting activity of neuronal subclasses measured with calcium dyes under various conditions: spontaneous (blue), evoked (NMA +5 HT, green), evoked + inhibitory antagonists (NMA +5 HT + strychnine + picrotoxin, purple), and evoked + glutamatergic antagonist (NMA +5 HT + CNQX, brown). (**R**) Spontaneous burst amplitude of V1 interneurons is lower than V3, V2a, and motor neuron circuitoids. Evoked burst amplitudes were unaffected by inhibitory antagonists with V3, V2a, and motor neuron networks. Mean ± SEM, {number} of independent differentiations and [number] of circuitoids indicated after each cell type. V3 interneurons (INs) {22}[175], V2a interneurons {5}[49], motor neurons (MNs) {11}[49], V1 interneurons {12}[42]. Unpaired t test: (ns) p>0.05; **p<0.01; ****p<0.0001. (**S**) V3 and V2a interneuron rhythmicity increased (I.C.V. decreased) under the evoked condition compared to spontaneous bursts. Mean ± SEM, {number} of independent differentiations and [number] of circuitoids indicated after each cell type. V3 {22}[175], V2a {5}[49], MN {11}[49], V1 {12}[42]. Paired t test: (ns) p>0.05; *p<0.05; ***p<0.001; ****p<0.0001. (**T**) Burst frequency increases in the evoked condition and decreases with

*Figure 4 continued on next page*

*Figure 4 continued*

glutamatergic antagonist CNQX. Mean ± SEM, {number} of independent differentiations and [number] of circuitoids indicated after each cell type. V3 {22}[175], V2a {5}[49], MN {11}[49], V1 {12}[42]. Paired t test: (ns) p>0.05; *p<0.05; ***p<0.001; ****p<0.0001.

The following figure supplements are available for figure 4:

**Figure supplement 1.** Cell purifications to generate synthetic neural networks.

**Figure supplement 2.** Frequency and rhythmicity of V3 interneuron networks.

**Figure supplement 3.** Cholinergic antagonists do not affect motor neuron networks.

FACS. Specific combinations of excitatory and inhibitory neuronal subtypes were plated onto an astrocyte layer (*Figure 5A*) where they formed interconnected networks that were spontaneously active. We found that excitatory V3 interneurons produced a reliable pattern of spontaneous bursts that continually increased in frequency when co-cultured with increasing numbers of inhibitory V1 interneurons (*Figure 5B and D*).

To test whether the inhibitory neurons entrained the bursting of V3 interneurons, we acutely blocked all inhibitory synaptic transmission in these cultures with picrotoxin and strychnine. After application of the inhibitory antagonists, the cultures returned to the bursting pattern observed without V1 neurons (*Figure 5B, D, F and G*). These findings reveal three aspects of the synthetic networks. First, the effect of inhibitory neurons is mediated by direct synaptic activity rather than causing changes in the development, survival, or physiological properties of V3 cells because picrotoxin and strychnine quickly restored V3 activity to its normal frequency. Second, the addition of V1 cells to V3 networks unlikely changes V3-activity by displacing cells or altering the connectome of the excitatory network because drug inhibitors of V1 interneurons restore normal V3-activity. Third, excitatory networks may tune their burst rate by flexibly recruiting different numbers of inhibitory neurons.

## Inhibition decouples motor neuron networks into separate units

Because networks comprised solely of purified motor neurons display different patterns of activity compared to V3 interneurons (see *Figure 4*), we next examined how increasing the ratio of inhibitory V1 interneurons influenced motor neuron activity (*Figure 5—figure supplement 1*). Unlike V3 networks, V1 interneurons had little influence on the burst frequency of motor neurons, but instead caused greater burst amplitude variation (*Figure 5C and E–G*).

To test whether the amplitude variability observed in motor neuron-V1 interneuron networks arose by converting the coordinated bursting of the network into subunits of desynchronized activity (termed segmentation) we refined how we monitored network output. Rather than measuring average neuronal bursts across the entire network we measured activity in sub-regions. In networks with a 10:1 motor neuron/V1 ratio, we found that separate regions of interest (ROI) could burst independently (*Figure 6A*, *Video 2*). Changes in the coordination of neuronal activity across the network were not due to depletion of excitatory interconnections among motor neurons or a physical alteration of the circuit's connectome, because inhibitory antagonists caused the network to rapidly switch back into a synchronous bursting pattern (*Figure 6B*, *Video 3*). The desynchronizing effect of inhibitory neurons on motor neuron networks was quantified as network complexity (*Figure 6C*, Materials and methods). Interestingly, inhibitory V1 interneurons lacked the ability to create complexity within V3 networks (*Figure 6C*, *Figure 6—figure supplement 1*).

To further explore the functional interactions between V1-V3 interneurons and V1-motor neurons, we used cell-type-specific reporters to monitor the activity of individual identified neurons within the mixed networks. In V1-motor neuron co-cultures we noted that motor neurons (GFP$^+$/Tomato$^-$) were co-active on some bursts but also had the freedom to fire independently (*Figure 6D*). When synaptic inhibition was blocked in these cultures all the motor neurons became synchronized (*Figure 6D and F*). In contrast, V3 interneurons always fired in synchrony within mixed V3-V1 cultures, however the frequency slowed in the presence of inhibitory antagonists (*Figure 6E and F*). Our results indicate that the cellular E/I ratio functions in a cell-type-dependent manner—namely, V1 interneurons

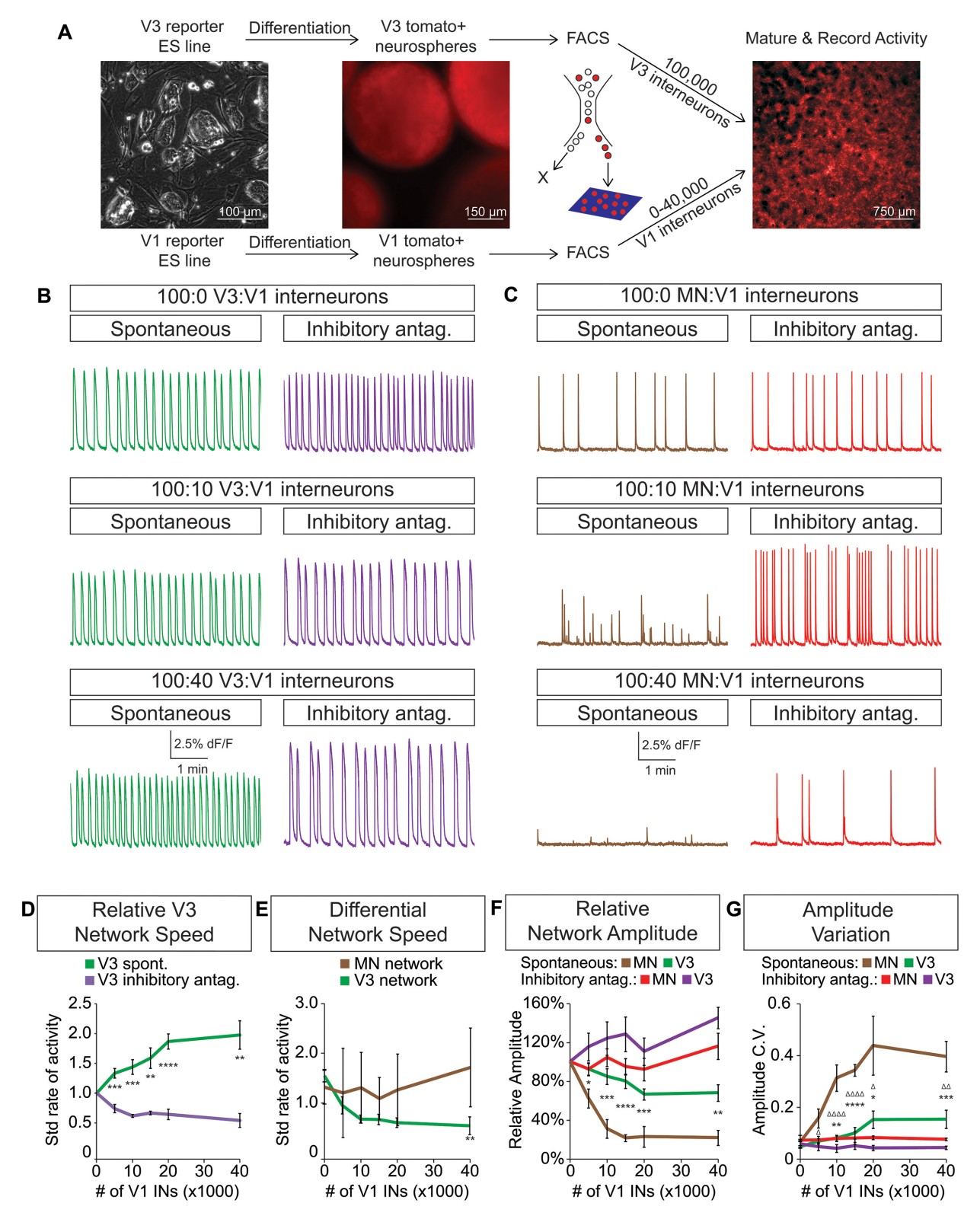

**Figure 5.** V3 network burst speed is tuned by inhibitory V1 interneurons. (**A**) ES cell lines with fluorescent reporters for neuronal subtypes were differentiated, and tomato +V3 and V1 interneurons were purified by FACS. Monolayer circuitoids were created by plating 100,000 V3 interneurons with 0–40,000 V1 cells onto astrocytes. Activity was recorded using calcium indicator dye to image co-culture networks 14 days after plating. (**B**) Spontaneous network bursting increases as the number of inhibitory V1 interneurons increases in the V3 interneuron cultures (green). Network frequency returns to

*Figure 5 continued on next page*

*Figure 5 continued*

baseline levels in the presence of inhibitory antagonists strychnine + picrotoxin (purple). **(C)** Motor neuron burst frequency does not increase as the concentration of V1 interneurons increases, but the amplitude of the bursts decreases (brown). Inhibitory antagonists strychnine + picrotoxin return motor neuron network burst amplitude to normal levels (red). **(D)** Quantification of burst frequency from traces in **(B)**. For each trial and condition, the standardized rate of activity was calculated by dividing the burst frequency of each network by the control network's (lacking V1 cells) rate. Mean ± SEM, n = 10 networks for each V1 concentration in the spontaneous condition, n = 4 networks with inhibitory antagonists for each V1 concentration. Unpaired t test: *p<0.05, **p<0.01, ***p<0.001, ****p<0.0001. **(E)** Quantification of relative burst rate from traces in **(B, C)**. Standard rate of activity was calculated by taking the ratio of a network's frequency in the inhibitory antagonist condition to its frequency in the spontaneous condition. Mean ± SEM, n = 6 motor neuron networks and n = 4 V3 interneuron networks at 0 and 40,000 V1 interneurons, unpaired t test: (ns) p=0.63; **p<0.01. **(F)** Quantification of relative burst amplitude from traces in **(B, C)**. Mean ± SEM, n = 10 V3 networks vs n = 7 MN networks for each V1 concentration tested, unpaired t test: **p<0.01; ***p<0.001; ****p<0.0001. **(G)** Amplitude coefficient of variation (C.V., Materials and methods) calculated from network activity in **(B, C)**. Motor neuron burst amplitude C.V. increased as V1 inhibitory cell number increased. Mean ± SEM, n networks in the (spontaneous) and [inhibitory antagonist] condition for V3 networks (10)[4] and MN networks (8)[≥4] at all V1 interneuron concentrations. MN spontaneous vs MN inhibitory antagonist (*) and MN spontaneous vs V3 spontaneous (Δ). Δ/*p<0.05; ΔΔ/**p<0.01; ***p<0.001; ΔΔΔΔ/****p<0.0001; unpaired t test.

The following figure supplement is available for figure 5:

**Figure supplement 1.** Generation of synthetic networks comprised of defined neuronal subtypes.

accelerate the rate of V3 interneuron bursts in a coordinated fashion across the entire network and uncouple the activity of motor neurons to establish separate units without changing overall burst frequency.

## Inhibitory V1 interneurons control motor neuron burst rate via V3 interneurons

Flexibility within motor behaviors requires the ability to dynamically switch the combination of motor pools that are co-active when synergistic muscles are recruited, while separately regulating the burst frequency to control speed. To understand how this might arise, we examined the activity of tripartite circuitoids comprised of motor neurons, V3 and V1 interneurons. Equal numbers of motor neurons and excitatory V3 interneurons were combined with increasing numbers of V1 inhibitory cells. We found that the tripartite circuitoids maintained their synchrony across the network regardless of V1 cell number (*Figure 7A–C*). The activity of tripartite networks resembled that observed for V3-V1 co-cultures studied above, as the relative bursting speed increased as V1 cells were added (*Figure 7D*). This prompted us to examine the activity of motor neurons within the tripartite networks using the GFP$^+$/Tomato$^-$ status of the cells to distinguish them from V1 and V3 interneurons in the culture. We found that within tripartite circuitoids motor neurons displayed the same coordinated pattern of activity as the interneurons and fired synchronously as a single unit within the culture across a range of V1 cell numbers (*Figure 7E–G*).

We considered the possibility that V3 interneurons can more strongly bind neurons within a network than motor neurons. To test this, we recorded miniature post-synaptic currents (mEPSCs) in networks composed of different neuronal subtypes. Within V3 networks, V3 interneurons displayed higher frequencies of miniature events than motor neurons in a pure motor neuron network (*Figure 7H,I,K*). Interestingly, the identified motor neurons in mixed V3-motor neuron circuitoids had similar mEPSC frequencies as the V3 interneurons in V3 networks (*Figure 7H–K*). This analysis indicates that the V3 cell-type may strongly bind the activity of neurons within a network due to a greater probability of neurotransmitter release or by providing an increased number of synaptic release sites.

## Discussion

Extensive work is underway to map and deconstruct in vivo circuits to understand how they function, complemented by in silico studies to model neural activity. Here, we have begun to bridge the two approaches by devising a system for modeling the control features of an oscillatory network using biological components. Specifically, we constructed oscillatory networks using defined neuronal subtypes generated from stem cells. We called these microphysical structures circuitoids, from which

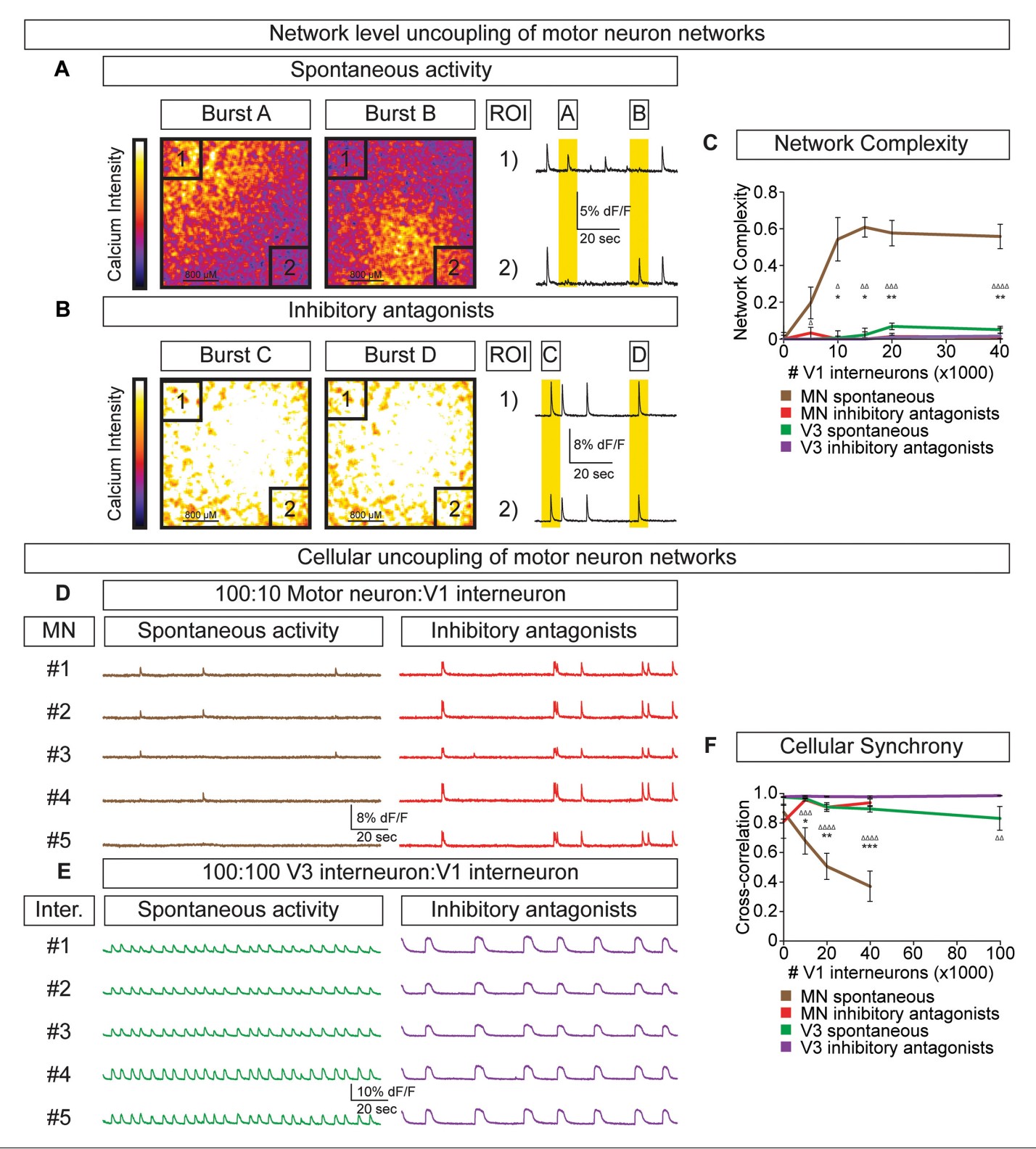

**Figure 6.** V1 interneurons control the segmentation of motor neuron network activity. Circuitoids were established with either 100,000 motor neurons or 100,000 V3 interneurons combined with 0–40,000 V1 interneurons, plated on astrocytes. Networks were imaged 14 days after sorting with a calcium indicator dye and the static frames show pseudocolored calcium intensity (dF/F) images (scale from black to white). (**A**) Neurons within plated circuitoids (10,000 V1 interneurons and 100,000 motor neurons) fire asynchronously. Static frames during Burst A and B reveal different areas of activity

*Figure 6 continued on next page*

*Figure 6 continued*

in the network. Traces from two regions of interest (ROI 1 and 2) on opposite corners of the field of view are displayed. (B) Inhibitory antagonists (strychnine + picrotoxin) applied to the network in (A) lead to synchronous bursts across the entire network. (C) Quantification of network complexity (Materials and methods) calculated from network activity. V1 interneurons increase network complexity of motor neuron networks. Median ± bootstrap standard error, n networks in the (spontaneous) and [inhibitory antagonist] condition for V3 networks (10)[4] and MN networks (8)[≥4] at all V1 interneuron concentrations. MN spontaneous vs MN inhibitory antagonist (*) and MN spontaneous vs V3 spontaneous (Δ). Δ/*p<0.05; ΔΔ/**p<0.01; ΔΔΔp<0.001; ΔΔΔΔp<0.0001 Kolmogorov–Smirnov test. (D) 100,000 purified GFP+ motor neurons were cocultured with 10,000 tomato +V1 inhibitory neurons. Calcium imaging was used to detect bursts from five individual GFP+ motor neurons. Individual cells displayed coupled activity (brown), but the entire cohort of motor neurons only became synchronously active when V1 inhibition was blocked with strychnine + picrotoxin (red). (E) 100,000 purified V3 interneurons were cocultured with 100,000 V1 cells, and five individual cells were calcium imaged. Individual interneurons consistently fired together whether V1 cells were active (green) or silenced with inhibitory antagonists (purple). (F) Quantification of neuronal synchrony using pair-wise cross-correlation analysis of motor neurons in motor neuron-V1 networks and interneurons in V3-V1 networks. The activity within motor neuron networks became increasingly less synchronized as V1 cell number increased. Mean±SEM, sample size n=(network number)[neuron number]. V3 network, spontaneous burst condition: 0–40,000 V1 cells (10)[200], 100,000 V1 cells (4)[80]. V3 network, inhibitory antagonist condition: 0–40,000 V1 cells (4)[80], 100,000 V1 cells (2)[40]. MN network, spontaneous burst condition, each V1 concentration tested (≥7)[≥140]. MN network, inhibitory antagonist condition, each V1 concentration tested (6)[120]. MN spontaneous vs MN inhibitory antagonist (*) and MN spontaneous vs V3 spontaneous (Δ) all compared at same V1 concentration, except for spontaneous 100,000 V1 in V3 network vs 40,000 V1 in MN network. Δ/*p<0.05; ΔΔ/**p<0.01; ΔΔΔp<0.001; ΔΔΔΔp<0.0001, unpaired t-test.

The following figure supplement is available for figure 6:

**Figure supplement 1.** Activity across different network configurations.

coordinated network oscillations emerged. To understand how flexible control might be imposed on the rhythmic networks, we studied the consequence of shifting the cellular E/I ratio on the bursting parameters. We found that inhibitory neurons act in a context-dependent manner to regulate the frequency of bursting and the pattern (i.e. segmentation) of the network activity. Our findings lead to a model whereby interconnected excitatory and inhibitory neurons can transition to different output based on cell-type-specific connection-strength hierarchies working in combination with controls that gate the relative activity-ratio of E-to-I cells (*Figure 8*).

## Networks that produce rhythmic activity

Rhythmic neuronal activity that is coordinated across circuits comprised of interconnected neurons is a general property found in many areas of the CNS (*Buzsaki, 2006*). This activity pattern probably arises when irregular spikes randomly produced by interconnected excitatory neurons become integrated across dendritic trees, thereby converging into a regular oscillatory pattern of output shared by cells across the network (*Softky and Koch, 1993*). This raises the paradoxical issue of understanding how diverse patterns of activity necessary for driving complex behaviors are produced by networks that default into stable patterns of rhythmicity. Theoretical considerations of this problem argue that inhibitory neurons are necessary in such systems (*Buzsaki, 2006*).

Although prevalent, the function of neural oscillations remains obscure in many cases. A wide range of frequency bands are observed in cortical structures ranging from 1 to 70 Hz and have been considered as possible substrates for binding information through synchronization and providing a time parameter to neural codes. Rhythmic neural activity likely plays roles in cognition and memory, and when unregulated produces tremors and seizures (*Buzsaki and Draguhn, 2004*; *Fries, 2005*; *Wang, 2010*).

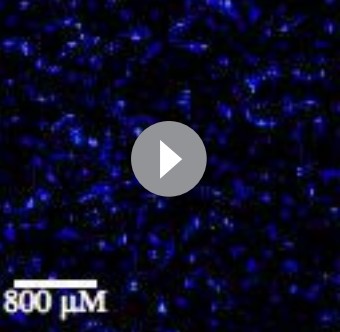

**Video 2.** V1 interneurons generate subnetwork activity in motor neuron-V1 networks. A network of 100,000 motor neurons and 10,000 V1 interneurons displays segmented activity. Calcium intensity change (dF/F) was pseudocolored (scale from black to white), showing different active regions within a network. Movie plays at 10x speed.

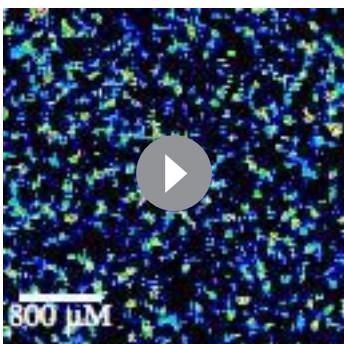

**Video 3.** Inhibitory antagonists synchronize motor neuron-V1 networks. The same network (see *Video 2*) displays synchronous network activity after the application of inhibitory antagonists (strychnine +picrotoxin), suggesting that synaptic activity from V1 inhibitory neurons patterns motor neuron networks. Calcium intensity change (dF/F) was pseudocolored (scale from black to white), showing coordinated activity across the network. Movie plays at 10x speed.

Aside from neural development (*Katz and Shatz, 1996*), the circuits where rhythmic activity plays the clearest role have central pattern generator attributes such as those that control repetitive movements linked to respiration, chewing, scratching, and walking for example.

The cellular components of the mouse spinal cord CPG have been investigated through functional studies using pharmacological agents and molecular tools to target distinct neuronal subclasses. Despite the extensive characterization of this circuit, it has remained unclear what drives the pacemaker system within the CPG. Several studies have found that perturbing the function of excitatory interneurons such as V2a or V3 cells degrades the rhythm. Likewise, we found that V3 interneurons are necessary for well-coordinated rhythmicity in heterogeneous networks. In addition, we found that V3 cells, V2a interneurons, and to a limited degree motor neurons were sufficient as isolated cell populations to establish rhythmic networks. These homogenous networks displayed somewhat regular spontaneous bursting patterns, but fired with increased frequency and higher regularity in drugs that evoke central pattern generator activity. It has been suggested that the pacemaker is an emergent property of interconnected excitatory neurons within the CPG, and that the default frequency is defined by the biophysical properties of the membrane, which cyclically convert cells from on-fire to off-fire states (*Grillner, 2006*; *Harris-Warrick, 2010*; *Kiehn et al., 2000*; *Marder and Bucher, 2001*). The inherent ability of a network to integrate neural properties across the cell population may represent one mechanism for helping to control for the intrinsic biophysical variability among neurons and thereby ensure a reliable and consistent output (*Marder et al., 2015*). Our findings support this view of pacemaker control and further suggest that robustness in the system is derived from the ability of multiple excitatory spinal cord cell types to default into regular oscillatory patterns.

## Circuitoids and central pattern generators

We generated subclasses of spinal-like neurons from ES cells harboring genetic reporters for specific neuronal types functionally implicated as components of the mouse spinal cord central pattern generator. Although the ES-cell-derived neuron subtypes used in this analysis without exception displayed features similar to their in vivo counterparts, we cannot exclude that there are differences between the in vitro and in vivo cells. The de novo generated neurons were either cultured as spheres or plated and allowed to form interconnections. Remarkably, after culturing we found that these synthetic networks produced rhythmic bursts of activity that resembled the activity of the spinal cord central pattern generator. Because these ES-cell-derived neural networks produced a behaviorally relevant activity pattern, we termed these microphysical systems circuitoids to distinguish them from neurospheres defined solely by their cellular composition. Pharmacological experiments and recordings of EPSCs indicated that circuitoid activity is synaptically driven, and that the synchronized bursting of the neurons comprising each circuitoid is likely a byproduct of the extensive interconnections among the cells. In addition to synaptic connections, gap junctions may also form among neurons in circuitoids. Nevertheless, antagonists of glutamatergic synapses disrupted bursting, indicating that gap junctions alone are insufficient to produce the oscillatory patterns of activity we observed. We found no evidence for migration of neurons into particular groupings or layers within the oscillatory-circuitoids in either aggregated or plated conditions; nor did we find evidence for a specific connectome among cells, other than cell-type-specific differences in network binding strength revealed in mEPSC recordings. Instead, the oscillatory activity we observed under a variety of conditions seemed to emerge simply from the intrinsic synaptic and membrane channel characteristics of the neuron subtypes and their ability to form extensive synaptic connections with one

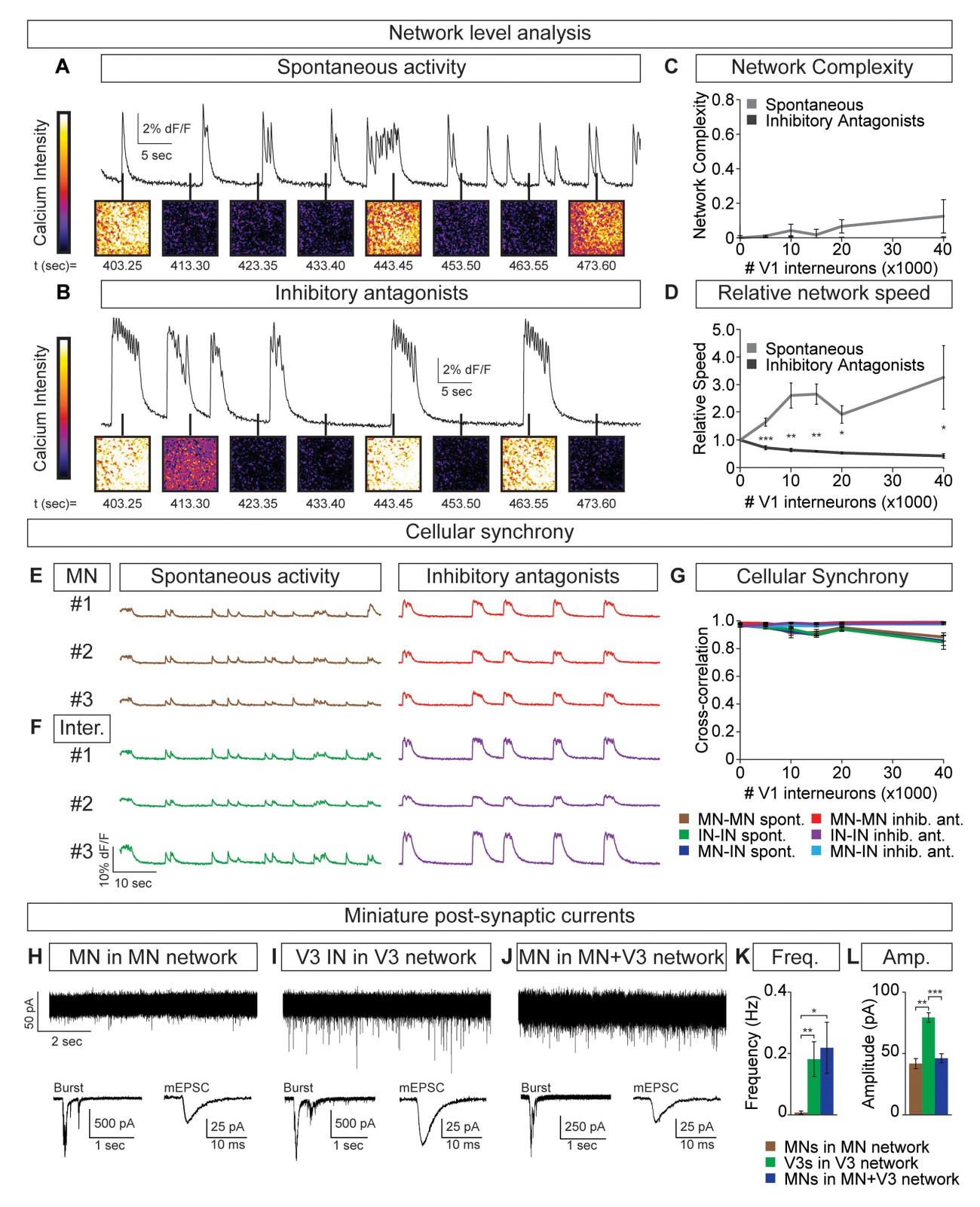

**Figure 7.** Motor neuron burst frequency is set by V3-V1 network activity. (**A–D**) Tripartite networks of 50,000 motor neurons, 50,000 V3 interneurons, and 0–40,000 V1 interneurons were formed, and burst activity monitored using calcium dyes (see *Figure 5A*). (**A**) Spontaneous activity of 50,000:50,000 V3-MN network with 40,000 V1 neurons. (**B**) Spontaneous activity of network in (**A**) with inhibitory antagonists (strychnine + picrotoxin). (**C**) Quantification of network complexity (Materials and methods) shows that unlike MN-V1 networks (see *Figure 6*) MN-V1-V3 networks burst synchronously. Median ±

*Figure 7 continued on next page*

Figure 7 continued

bootstrap standard error, n = 5 for each V1 concentration and condition. (D) Increasing numbers of V1 interneurons in V3-MN networks increases burst frequency. Inhibitory antagonists reduce burst frequency of V1-V3-MN networks. For each trial, the frequency of the networks in each condition was standardized to the burst rate of control V3-MN networks (lacking V1 cells). Mean ± SEM, n = 5 for each V1 concentration and condition. Paired t test: *p<0.05, **p<0.01, ***p<0.001. (E–G) Burst analysis of individual fluorescent-labeled cell types in V1-V3-MN networks prepared as described in (A–D). (E) Motor neuron bursting in a 50,000:50,000 V3-MN network with 40,000 V1 neurons. Spontaneous bursts (brown) and activity with inhibitory antagonists (strychnine + picrotoxin, red) are shown. (F) Interneuron bursting in the same network as (E). Spontaneous burst (green) and activity with inhibitory antagonists (purple) are shown. (G) Neuronal synchrony quantification using pair-wise cross-correlation analysis of MN-MN, IN-IN, and MN-IN activity. The activity of each neuronal combination is highly correlated in V1-V3-MN networks. Mean±SEM, sample size n=(network number)[neuron number] (5)[50] for all cell populations and conditions at each V1 concentration. (H–L) Recordings of miniature post-synaptic currents (mEPSC) in 0.5 μM TTX. (H) Synaptic activity in pure motor neuron network. (I) Synaptic activity in pure V3 interneuron network. (J) Motor neuron mEPSCs in MN-V3 network. (H-J) Insets show synaptic drive associated with network bursting prior to TTX application (Burst, left) and averaged miniature events from one neuron (mEPSC, right). (K) Frequency of mEPSCs is reduced in pure motor neuron networks. (L) Amplitudes of motor neuron mEPSCs are smaller than V3 interneurons. Mean ± SEM, n = 6 MNs from MN networks; n = 4 V3 INs from V3 networks; n = 6 MNs from MN-V3 networks. *p<0.05; **p<0.01; ***p<0.001.

another. Our findings indicate that the coordinated oscillatory behavior of a rhythmic network can arise without a stringent requirement for a particular circuit architecture that relies upon a specific physical arrangement of pre- and post-synaptic cells. Interestingly, motor neurons acquire rhythmic activity even when ectopically located within the spinal cord (*Hinckley et al., 2015*; *Machado et al., 2015*).

Can circuitoids comprised of spinal-like neurons be compared to spinal cord central pattern generators? The oscillatory networks studied in this report lacked the structured inhibition that underlies left-right and flexor-extensor coordination found in complex CPG networks such as the lumbar spinal cord (*Kullander et al., 2003*; *Zhang et al., 2014*). Rather, the oscillatory activity of circuitoid networks was more akin to a half-center CPG or respiratory CPG, which produce regular on-off bursts (*Garcia-Campmany et al., 2010*). The similarities between circuitoids and simple CPGs were several fold. (1) Circuitoid oscillations were synaptically driven by glutamate, like CPGs (*Beato et al., 1997*; *Hägglund et al., 2010*; *Kiehn et al., 2000*). (2) Circuitoids displayed spontaneous activity that resembled the spontaneous bursts recorded from isolated spinal cords (*Myers et al., 2005*; *O'Donovan and Landmesser, 1987*; *Whelan et al., 2000*). (3) Circuitoids became rhythmically active in drugs that evoke CPG activity (*Jiang et al., 1999*; *Kudo and Yamada, 1987*; *Smith and Feldman, 1987*; *Whelan et al., 2000*). (4) The frequency of bursts produced by circuitoids was in a similar range to those produced by the spinal CPG (*Talpalar and Kiehn, 2010*; *Whelan et al., 2000*). (5) The ablation of V3 interneurons from circuitoids degraded the rhythmicity analogous to the phenotype caused by ablating V3 cells in vivo (*Zhang et al., 2008*). (6) Like spinal CPGs, circuitoid rhythms could occur in the absence of inhibition (*Bracci et al., 1996*; *Cowley and Schmidt, 1995*; *Kremer and Lev-Tov, 1997*). (7) In both circuitoids and CPGs blocking inhibitory V1 interneuron function reduced the burst speed (*Cowley and Schmidt, 1995*; *Gosgnach et al., 2006*).

Although there are likely important differences between circuitoids produced in vitro from ES cells and CPGs, the ability to create networks with defined cell types and numbers represents a unique approach for testing the sufficiency and instructive qualities of cell types for particular activity patterns. Functional studies of neurons within oscillatory circuits have been performed by titrating pharmacological agents; however, this global alteration of synaptic signaling strength may not be comparable to the way different cell combinations are dynamically selected for activity within a network by higher control centers. Consequently, we designed our experiments around using defined cell mixtures to study network dynamics, rather than using drug titrations, allowing us to have tight control over the cellular E/I makeup of networks, an aspect that cannot currently be controlled as precisely with in vivo manipulation.

## A cellular E/I model to produce flexible motor behaviors

While the underpinnings of circuit-switches are poorly understood, rhythmic networks within the spinal cord are clearly capable of dynamically changing the speed and pattern of motor neuron activation to produce an elaborate repertoire of motor actions. Functional studies have established inhibitory neurons as necessary components of the circuitry that patterns motor output (*Cowley and*

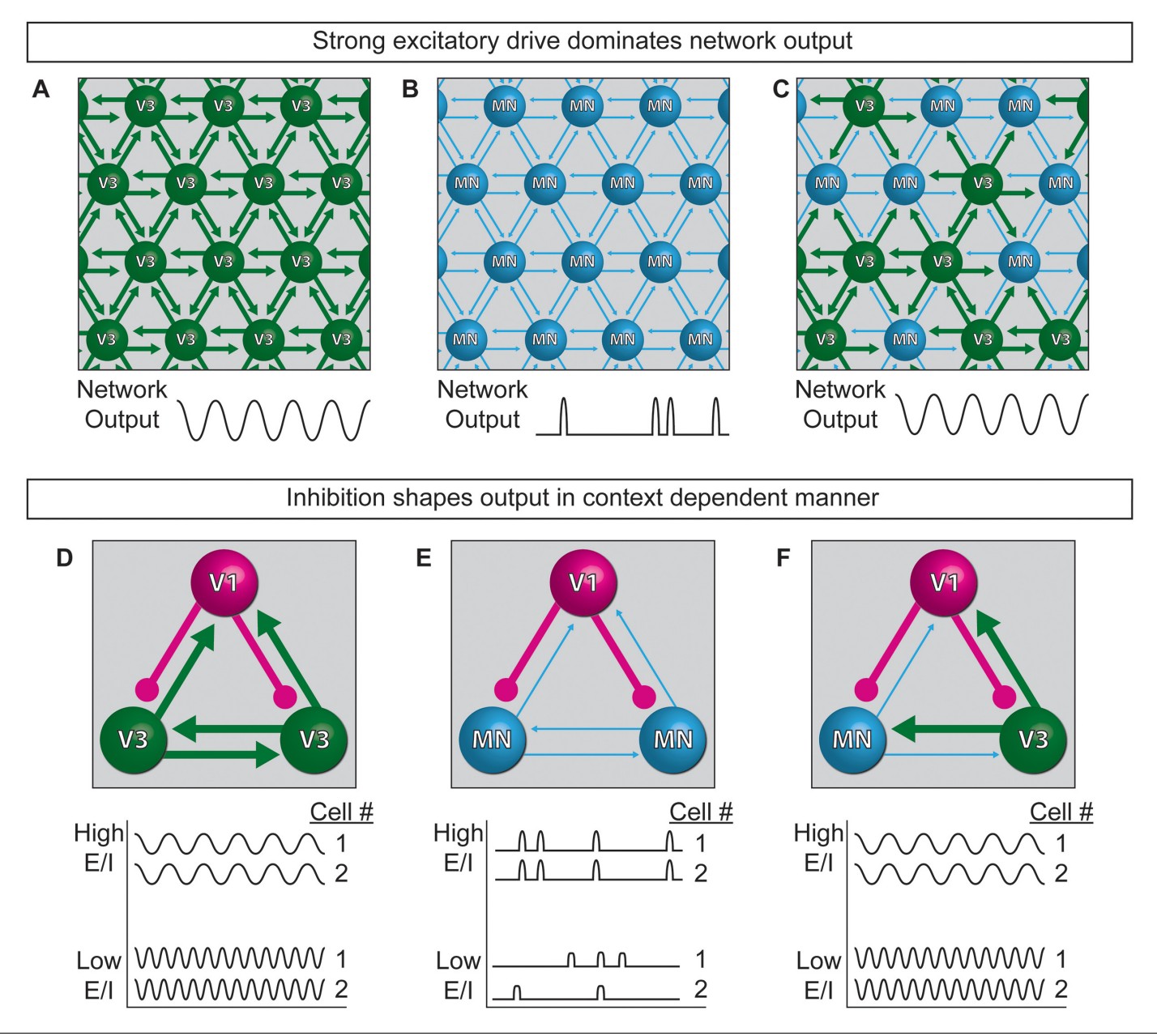

**Figure 8.** Model of neuronal interactions in an oscillatory circuit. (**A–C**) V3 interneurons form strong synaptic interactions, whereas motor neurons have weaker connections. (**A**) Strongly connected neurons in V3 networks become rhythmically active. (**B**) Weakly connected neurons in MN networks burst with less regularity. (**C**) V3 interneurons bind MNs into a network with stronger inter-connections, leading to more regular network activity. (**D–F**) V1 inhibitory neurons shape the activity of V3 and MN cells differently. (**D**) The E/I ratio established between V1-V3 interneurons influences burst frequency. This is analogous to speed regulation of oscillators such as CPGs. (**E**) The E/I ratio established between V1-MN neurons influences the pattern of bursts across the network. This is analogous to switching the coupling and uncoupling of activity among different motor pools during motor behaviors. (**F**) In networks with a mixture of V1-V3-MN cells, the specific combination of E/I balance and degree of interconnectedness among networked neurons are predicted to influence the burst speed and segmentation of activity.

*Schmidt, 1995*; *Gosgnach et al., 2006*); however, it is not possible to determine if particular neuron subtypes are instructive components involved in CPG flexibility by eliminating a cell type from the system. To explore the consequences of changing the E/I ratio within oscillatory circuits, we made synthetic networks with identified neuronal subtypes mixed in controlled numbers. Base networks of

purified excitatory neurons with strong interconnections, such as those comprised of V3 interneurons, reliably produced regular rhythmic bursts (*Figure 8A*). Networks of weakly interconnected excitatory neurons, such as motor neurons, burst with more irregular patterns, but adopted a regular synchronized activity-pattern in the presence of interneurons with strong connections (*Figure 8B and C*). Thus, under the cell mixing conditions we tested, neurons with the intrinsic ability to form strong connections establish dominance in the network.

One manner in which different behaviors might emerge from a highly interconnected network like the spinal cord, is by descending inputs recruiting different proportions of inhibitory and excitatory neurons within the network and thereby changing the E/I balance. Although we could not mimic the structured recruitment of cell types by descending inputs, we could investigate the consequence of changing the E/I balance of networks by mixing different combinations of excitatory and inhibitory neuron subtypes. V1 interneurons increased the burst frequency of strongly interconnected V3 networks in proportion to the E/I cell ratio (*Figure 8D*), suggesting V1 interneuron influence on network speed can occur upstream of motor neurons. When V1 inhibitory neurons were combined with weakly connected motor neurons they had little influence on burst frequency, but they caused subnetworks to emerge in proportion to the E/I cell ratio (*Figure 8E*). Although our findings do not explain how the activity of different motor pools is coordinated, our results support the possibility that the selective recruitment of increasing numbers of V1 cells could switch motor pools from a coupled (synchronous) to an uncoupled (segmented) state in order to facilitate the differential recruitment of muscles during complex behaviors (*Figure 8E*). In tripartite circuits with weakly- and strongly-connected excitatory neurons in combination with inhibitory neurons, we found that the entire network adopted a synchronous pattern of activity whose frequency increased as the number of inhibitory neurons increased (*Figure 8F*). The selective recruitment of V1 cells into the CPG may be facilitated by the large genetic diversity that has been uncovered within this cell group (*Bikoff et al., 2016*; *Francius et al., 2013*).

Our observations suggest that flexibility within rhythmic circuits can be achieved by constructing networks with neurons that have unequal synaptic interactions, and by regulating the activity-ratio of excitatory and inhibitory neurons. Together these features are sufficient to control the speed and segmentation of bursting. It has not escaped our notice that circuitoids could be used to study neurological diseases that affect circuits and might represent transplantable modules for nervous system repair.

## Material and methods

All experiments were conducted in accordance with the Salk Institutional Animal Care and Use Committee Animal Protocols (IACUC #11–00020) and the Salk Institutional Review Board (IRB).

### Mouse and embryonic stem cell lines

The generation and genotyping of the En1:Cre, Chx10:Cre (Chx10 is encoded by the *Vsx2* gene), Hb9:GFP, and Sim1:Cre alleles in mice has previously been described (*Azim et al., 2014*; *Gosgnach et al., 2006*; *Lee et al., 2004*; *Sapir et al., 2004*; *Zhang et al., 2008*). The Gt(ROSA) 26Sor^tm9(CAG-tdTomato) (R26/C:LSL:Tomato) and Gt(ROSA)26Sor^tm1(DTA) (R26:LSL:DTA) lines were obtained from Jackson Laboratory (007905 and 010527, respectively). Transgenic CAG:GCaMP3 mice were generated by using restriction enzymes to cleave the promoter+reporter fragments from the bacterial plasmid, and injecting the purified DNA into mouse oocyte pronuclei. After microinjection, founders were genotyped by PCR with the GFP primers and screened for ubiquitous presence of GCaMP3.

All ES cell lines were derived as novel lines for the experiments in this paper. Blastocysts were flushed 3.5 days after fertilization using M2 media (MR-015-D, Millipore). Each individual blastocyst is placed in one well of a 96-well plate containing primary mouse embryonic fibroblasts (pMEF - GlobalStem) with 2i media (SF016-100, Millipore). After 5 days of incubation, the 2i media is aspirated and each hatched blastocyst is dissociated using accutase and passaged to one well of a 24-well plate with pMEF and 2i media. Colonies are visible after 1 or 2 days. Every second passage with accutase decreases the concentration of 2i media from 100% to 75%, 50%, 25% and finally to 0% with FCS media [Knockout DMEM (Life Technologies, now Thermofischer Scientific, Waltham MA), 1X HEPES (Life Technologies), 1X non-essential amino acids (Life Technologies), 200 mM L-glutamine

(Life Technologies), 10% ES-qualified fetal bovine serum (Millipore), 0.1 mM $\beta$-mercaptoethanol (Sigma), 1,000–2,000 units of leukemia inhibitory factor (LIF) (Calbiochem), 1X Antibiotic-Antimycotic (Life Technologies)] making up the other fraction. After colonies were established, ES cells were passaged as needed using 0.25% trypsin (Life Technologies) and plated into FCS media. At times, 2x the concentration of LIF was used to improve ES cell colony morphology. All ES cell lines were genotyped by Transnetyx using the same protocols to genotype the mouse lines from which they were derived. All lines were negative for mycoplasma contamination, as verified with a PCR screen.

## Differentiation of embryonic stem cells

ES cells are differentiated in suspension in 10 cm petri dishes. $1 \times 10^6$ dissociated ES cells are resuspended in 10 ml ADFNK media [Advanced D-MEM/F-12 (Life Technologies): Neurobasal medium (Life Technologies) (1:1), 10% Knockout Serum Replacement (Life Technologies), 200 mM L-Glutamine (Life Technologies), and 0.1 mM $\beta$-mercaptoethanol (Sigma)]. Two days later, embryoid bodies (EBs) were allowed to settle to the bottom of a 15-mL conical tube. Media was aspirated, and a third to a tenth of the EBs were transferred to a new 10-cm plate with fresh ADFNK media that was supplemented with 1 µM all-trans retinoic acid (RA, Sigma) and 5 nM to 1000 nM smoothened agonist (SAG, Calbiochem). Two days later, freshly supplemented media was exchanged (*Peljto et al., 2010*; *Wichterle and Peljto, 2008*; *Wichterle et al., 2002*). For DAPT application, following the 6 days of differentiation, 5 µM *N*-[*N*-(3,5-difluorophenacetyl-I-alanyl)]-(*S*)-phenylglycine t-butyl ester (DAPT; Sigma), a Notch inhibitor, was applied for four days prior to FACS. On day 6, if to be used for sorting, heterogeneous neurospheres were maintained in non-supplemented ADFNK media. To maximize sorting efficiency (greatest fluorescent+ population and ease of dissociation) for generation of pure or mixed circuitoids through FACS, Hb9:GFP ES cell lines were sorted on days 6–7 and all Cre-dependent tomato+ lines were sorted on days 10–11. If used for imaging, heterogeneous neurospheres were switched to a neuronal media [Neurobasal medium (Life Technologies), 2% ES-qualified fetal bovine serum (Millipore), 200 mM L-Glutamine (Life Technologies), 1X B-27 supplement (Life Technologies), L-glutamic acid (Sigma), 1X Antibiotic-Antimycotic (Life Technologies), 10 ng/ml Human Brain Derived Neurotrophic Factor (BDNF, Peprotech 450–02) and 10 ng/ml Recombinant Murine Glial-Derived Neurotrophic Factor (GDNF, Peprotech 450–44)]. Half the media was exchanged three times a week until activity was recorded. Activity of these heterogeneous networks, unless otherwise noted, was recorded 15–17 days from ES cells.

## Fluorescent-activated cell sorting

Neurospheres, 6–11 days from ES cells, were dissociated (Papain, Worthington), and then counted with a BD FACScan to determine the percentage of neurons composing neurospheres at different SAG concentrations. For sorting and generating purified networks, the BD FACSDIVA and BD Influx were used to sort neurons into low-adherent, u-bottomed 96-well dishes (Corning 7007).

## Astrocyte preparation

To derive cortical astrocytes for reaggregated de novo networks we used a similar protocol to McCarthy and DeVellis procedure described previously, adapted for mice (*Ullian et al., 2001*). P0-P3 mouse cortices were dissected and dissociated (Papain, Worthington). They were grown in a T-75 flask for 3–4 days with AGM media [DMEM + GlutaMAX (Life Technologies), 10% ES-qualified fetal bovine serum (Millipore), 1 mM Na-Pyruvate (Life Technology), 5 µg/ml insulin (Sigma I1882), 5 µg/ml n-acetylcysteine, 1 µM hydrocortisone (Sigma – H0888), 1X Antibiotic-Antimycotic (Life Technologies)]. Following a 1X PBS wash, contaminating cells were shook off and then the media was exchanged. 1–3 days later, once confluent, 10 µM AraC (Sigma 1768) was added and subsequently removed 2 days later with three washes with 1x PBS. To use, cells were dissociated in 0.05% Trypsin (Life Technologies). For astrosphere formation 50,000 astrocytes were placed in each well of a 96-well ultra-low adherent u-bottomed plate (Corning – 7007) and spun at 300 g to aggregate the cells. If compact spheres were not formed 24 hr later, light trituration was used to break the aggregate apart, and then the plate was respun. For plated assays, 100,000 astrocytes per 100–200 µl were plated onto the glass coverslip of the 35 mm dish (Corning 354077) for 2 hr. After cell adhesion, an addition 2 ml of AGM media was added to the dish. At times, to aid in adherence, prior to plating the astrocytes, dishes were recoated with 10 µg/ml poly-d-lysine (Sigma P6407) or 10 µg/ml laminin

(Life Tech 23017–015). Neurons were plated onto astrospheres or confluent astrocytes 2–4 days after astrocyte dissociation.

## Generation of highly defined de novo networks

To generate reaggregated circuitoids, differentiated heterogeneous neurospheres were dissociated between days 6 and 11 and specific neuronal subtypes were sorted directly into a well of a 96-well ultra-low adherent u-bottomed plate (Corning 7007). After sorting, these plates were spun to pellet the neurons. FACS sheath was removed without disrupting the loose pellet at the bottom of the well and neuronal media was added back. two more washes were conducted, combining wells if necessary (some experiments would have caused wells to overflow during FACS if neurons for one network were not split between multiple wells). For sphere assays, the neurons were resuspended and transferred into a well of a u-bottomed 96-well plate that was already filled with an astrosphere. These plates were now spun to increase contact of neurons and astrosphere. The following day, light trituration was used to remove any debris from the main reaggregated sphere. A few smaller satellite spheres may have also formed, so the plate was spun once more. 24 hr later, all the neurons and astrocytes formed one coherent sphere. All the media were carefully removed from the cells and new media was added. Unless otherwise noted, the circuitoids were incubated for 3 weeks after FACS before their network activity was imaged, with half of their media being exchanged three times a week. For plated networks, the AGM from the 35 mm dish was aspirated sufficiently to dry the plastic surrounding the 10 mm coverslip, and then media on the coverslip was aspirated. Neurons for each network were resuspended in 100–200 µl of neuronal media and plated onto the confluent astrocyte layer. After 2 hr the neurons had adhered and 2 ml of neuronal media were added to the 35 mm dish. The plated networks were incubated for 2 weeks after FACS before their network activity was imaged, with half of their media being exchanged three times a week.

## Optical and electrical recordings

During recordings, samples were perfused in ACSF (128 mM NaCl; 4 mM KCl; 21 mM NaHCO$_3$; 0.5 mM NaH$_2$PO$_4$; 1 mM MgSO$_4$; 30 mM D-glucose; and 2 mM CaCl$_2$) bubbled with a 95/5/% O2/CO2 mixture. Calcium signal was recorded using either a ubiquitously expressed GCaMP3 (tgCAG: GCaMP3) or Oregon Green 488 BAPTA-1-AM (Life Technologies). For dye application, Oregon Green 488 BAPTA-1-AM (Life Technologies) was applied at 10 µM in ACSF for 1 hr in a 37°C incubator. Dye was washed out with perfusion of ACSF for 15 min prior to recording. Unless otherwise stated, image series were acquired on an upright epifluorescent Olympus microscope (BX51WI) using a Hamamatsu C9100-13 camera and ImageJ plugin: µManager software (RRID:SCR_000415), capturing 20 frames/second at 128 × 128 using a 4 × 0.28 NA air objective (Olympus) with a 0.63x camera mount, and an X-Cite exacte light source at 2% power. For *Figures 6D–F and* and *7E–G* cellular resolution data of plated networks was acquired at 512 × 512 using a 20 × 1.0 NA water-immersion objective (Olympus) with a 0.63x camera mount.

For electrical, population recordings, activity was measured using a suction electrode with a multiclamp 700B amplifier, filtered at 300 Hz to 1 kHz. For whole-cell patch clamping, cells were visualized for whole-cell patch recordings using a BX51WI (Olympus) microscope equipped to allow both differential interference contrast (DIC) and epifluorescence imaging. Recordings were performed using a Multiclamp 700B amplifier (Molecular Devices). Signals were filtered at 6 kHz, sampled at 50 kHz through an Axon Digidata 1550A interface device (Molecular Devices) and recorded using Clampex 10 software (Molecular Devices). Electrodes were pulled using a P-97 flaming-browning micropipette puller (Sutter Instruments) from thick-walled borosilicate glass GC150F capillaries (Harvard Apparatus) to a resistance of 2–5 MΩ. All cells were voltage-clamped at −60 mV. Series resistance of 4–10 MΩ was compensated by 20–40% and recordings were abandoned if it increased by more than 20%.

During recording, cell cultures were continuously perfused at 5–8 ml/min with ACSF bubbled with a 95/5/% O$_2$/CO$_2$ mixture. The pipette solution consisted of (in mM) 140 Cs-gluconate, 4 CsCl, 2 CaCl2, 10 HEPES, 5 EGTA, 2 MgATP, 3 QX-315 Br, pH 7.3 with CsOH, and osmolarity of 290–310 mOsm. Burst activity was recorded in the first cell of each cell culture prior to the wash in of 500 nM TTX. Miniature post-synaptic potentials were detected using WinEDR 3.2.4 (Strathclyde

Electrophysiology Software) and analyzed using Clampfit 10.2 (Molecular Devices) with further analysis completed in R.

## Pharmacology of network activity

Drugs used were in the following final concentrations: 20 µM N-Methyl-DL-aspartic acid (NMA Sigma M2137 – discontinued, 10 µM NMDA M3262 equivalent), 40 µM serotonin creatinine sulfate monohydrate (5-HT, Sigma H7752), 50 µM dihydro-$\beta$-erythroidine hydrobromide (DH$\beta$E, Tocris 2349), 10 µM mecamylamine hydrochloride (MLA, Tocris 2843), 10 µM CNQX disodium salt (Tocris 1045), 1 µM strychnine hydrochloride (Sigma S8753), and 10 µM picrotoxin (Sigma P1675), 500 nM tetrodotoxin citrate (TTX, Cayman Chemicals 14964).

## Immunohistochemistry

Neurospheres and circuitoids were fixed in 4% PFA for an hour and then washed in 1x PBS and prepared for cryosectioning. Cryosections (60 µm) were stained in 1x PBS containing 1% BSA and 0.1% triton with NeuroTrace 640/660 Deep-Red Fluorescent Nissl Stain (Life Technologies N-21483, RRID: AB_2572212), DAPI, rabbit anti-PSD95 (1:1000, Life Technologies, RRID:AB_2533914), guinea pig anti-VGLUT2 (1:3000, Millipore, RRID:AB_1587626), goat anti-dsRed (1:500, Santa Cruz), rat anti-RFP (1:1000, Chromotek, RRID:AB_2336064), and rabbit anti-GFAP (1:500, Dako, RRID:AB_10013482). Imaging was conducted on an Olympus FV1000 confocal.

## RNA isolation

Days 6–8 neurospheres and e12.5 spinal cords, micro-dissected (Leica stereomicroscope) from mice, were dissociated with papain (papain dissociation kit, Worthington Biochemical). After 45 min, tissue was triturated and centrifuged at 1000 rpm for 5 min. Dissociated cells were resuspended in 1:1 Neurobasal:DMEM/F12 (without phenol red) with 3% Horse Serum (Invitrogen) and DNase (Worthington Biochemical). Before sorting, cells were passed through a 35-µm cell strainer (08-771-23, BD Falcon). Sorting was conducted on a Becton Dickinson FACS Vantage SE DiVa using Coherent Sapphire 488 nm and 568 nm solid state lasers (200 mW). Cells were collected directly into miRvana RNA lysis buffer and stored at −80C. RNA was isolated using the miRvana miRNA isolation kit (Ambion AM1560) or RNeasy Mini Kit (Qiagen). Each in vivo sample is a pool of isolated cells from one to three spinal cords, as necessary, to obtain sufficient RNA (quantified by Agilent TapeStation) for RNA sequencing.

mRNA sequencing libraries were prepared using the Illumina TruSeq RNA Library Preparation Kit (v2) according to the manufacturer's instructions. Briefly, RNA with polyA+ tails was selected using oligo-dT beads. Then, mRNA was fragmented and reverse-transcribed into cDNA. cDNA was end-repaired, index adapter-ligated and PCR amplified. Nucleic acids were purified with AMPure XP beads (Beckman Coulter) after each step. Libraries were then quantified, pooled, and sequenced using either the Illumina HiSeq 2500 or Illumina HiSeq 2000 platforms at the Salk NGS Core and Beijing Genomics Institute. Sequencing libraries were either 50 bp single-end or 100 bp paired-end.

## Gene expression quantification

To help control library type bias in the quantifications, raw FASTQ reads were trimmed to 50 bp and processed as single-end. Reads with <15 minimum average base quality were filtered out. Trimmed and filtered reads were quantified using kallisto against the most recent release of the Refgene annotation for mm10 (downloaded from UCSC Genome Browser) (*Bray et al., 2016*). Gene level TPM expression values were obtained by summing isoform level TPM at each gene loci.

## Data analysis

Data were exported from ImageJ (RRID:SCR_003070) using the ROI manager after manually drawing regions of interest around spheres, or for plated networks, ROIs were drawn around individual neurons (cellular analysis), or the full or partial field of view (network analysis). Burst detection was carried out in Igor Pro (RRID:SCR_000325) with TaroTools, Dr Taro Ishikawa, (https://sites.google.com/site/tarotoolsregister) or a custom pipeline generated in R.

To analyze network complexity, movies were converted to 8 bit grey-scale and resized to 128 × 128 pixels and exported as TIFF stacks from ImageJ. TIFF stacks were imported into R for

analysis. Stacks were transformed into 196 'signals' where each signal is the averaged intensity vs time from an 11 × 11 pixel oval shaped ROI. ROIs were taken at nine pixel spacing to allow for a small amount of overlap between adjacent ROIs. The size of the ROIs was selected manually based on training data to maintain good detail while still averaging out imaging noise. Signals were smoothed with a 1 s wide running mean filter and delta-f / f transformed. The full set of ROIs from a movie was then PCA transformed. The principal components were separated into 'signal' and 'noise' using the scree-plot method of significant component estimation (finding the component number at the elbow of the plot). Based on training data, we occasionally found that information was lost by selecting the component at the elbow of the scree-plot so, instead, selected two past the elbow. The significant components were recombined to form the signal matrix and all non-significant components were recombined to form the noise matrix. The sum of these two matrices is equal to the original input to PCA. For each signal, the standard deviation of the noise version was used to threshold the signal version for burst location calling. Burst locations were identified as rising edges in the signal version of an ROI with change in intensity greater than approximately 1.96 times the standard deviation of the noise version of the same ROI. This threshold was selected manually based on training data and gave a good balance between false-positive and false-negative burst calling. Change in intensity was measured from the base to peak of the rising edge of the burst.

Burst positions from all ROIs within a movie were cataloged to create a set of distinct burst positions within ±1 s windows (the burst catalog). Each ROI was then re-described as a sequence of the cataloged bursts. We then filtered out bursts from the catalog that appeared in less than 4% of the ROIs. This threshold was set manually based on training data to balance out Type I and Type II error as fair as possible and then used for all primary analysis. After filtering, the ROIs were re-described again. This re-described set of ROIs represents all robust bursting activity within a single movie. To quantify the complexity, taken as a deviation from the state of all ROIs having identical activity, we built a directed graph of the burst sequences within a movie. Briefly, a graph is a type of data structure with one or more nodes connected by edges. They are often used to describe networks. If the edges are assigned a direction then the graph is a directed graph. In our case, the nodes are the cataloged bursts. Edges were added to the graph between any two bursts that occurred adjacently in time in any ROI. The direction of the edge is in the direction of occurrence in time. If, in one ROI, burst Y followed burst X and was adjacent to burst X (no bursts occurred between them) then an edge would be added connecting from X to Y. From the point of view of a node in the graph, edges may be incoming or outgoing. The foregoing connection of X to Y has a single edge. From the point of view of node X it is an outgoing edge while from the point of view of Y it is an incoming edge. Complexity of a movie was then defined as the average of both the ratio of nodes with more than one outgoing edge and ratio of nodes with more than one incoming edge. If all ROIs in a movie have the same burst sequence then, in terms of the graph, each node would only be connected to one other node resulting in a baseline score of 1 for the movie. When the ROIs share some bursts, but not all, then the nodes of the graph begin to have more than one incoming or outgoing edge, which increases the final complexity score. We did not consider a case for ROIs that do not share any bursts because we did not observe any such cases in our data. The complexity statistic was found to be non-normal across the different testing groups and to have unequal variances so the two-sample Kolmogorov-Smirnov test was used to perform statistical tests between groups. All other data were compared using a student t-test.

## Acknowledgements

C Dowling and N Allen provided valuable guidance on astrocyte preparations. We thank A Levine, C Stevens, and K Lettieri for support and advice. This work was supported by the Flow Cytometry Core Facility of the Salk Institute with funding from NIH-NCI CCSG: P30 014195 with direct help from C Fitzpatrick and C O'Connor. MJS was supported by the Rose Hills Foundation, the HA and Mary K Chapman Charitable Trust, and the UCSD Neurosciences Graduate Program Training Grant. CAH was supported by a US National Research Service Award fellowship from US National Institutes of Health NINDS. KLH was supported as a National Science Foundation Graduate Research Fellow. MH was supported by the Japanese Ministry of Education, Culture, Sports, Science, and Technology Long-Term Student Support Program and the Timken-Sturgis Foundation. NDA was supported by NINDS fellowship (F31-NS080340-03). WDG was supported by CIRM. SLP is an HHMI investigator

and Benjamin H Lewis chair in neuroscience. This research was supported by the Howard Hughes Medical Institute, the Christopher and Dana Reeve Foundation, the Marshall Heritage Foundation, and the Sol Goldman Charitable Trust.

## Additional information

### Funding

| Funder | Grant reference number | Author |
| --- | --- | --- |
| Christopher and Dana Reeve Foundation | | Matthew J Sternfeld<br>Niall J Moore<br>Matthew T Pankratz<br>Samuel L Pfaff |
| Mary K. Chapman Foundation | | Matthew J Sternfeld |
| Rose Hills Foundation | | Matthew J Sternfeld |
| University of California San Diego | Neuroscience Graduate Program Training Grant | Matthew J Sternfeld |
| Howard Hughes Medical Institute | | Christopher A Hinckley<br>Niall J Moore<br>Matthew T Pankratz<br>Shawn P Driscoll<br>Dario Bonanomi<br>Samuel L Pfaff |
| National Institute of Neurological Disorders and Stroke | | Christopher A Hinckley |
| US National Research Service Award | | Christopher A Hinckley |
| National Science Foundation | | Kathryn L Hilde |
| Ministry of Education, Culture, Sports, Science and Technology | Long-Term Student Support Program | Marito Hayashi |
| Timken-Sturgis Foundation | | Marito Hayashi |
| National Institute of Neurological Disorders and Stroke | F31-NS080340-03 | Neal D Amin |
| California Institute for Regenerative Medicine | | Wesley D Gifford<br>Samuel L Pfaff |
| National Institute of Neurological Disorders and Stroke | NS090919 | Martyn Goulding |
| Benjamin H. Lewis Chair | | Samuel L Pfaff |
| National Cancer Institute | CCSG: P30 014195 | Samuel L Pfaff |
| Marshall Heritage Foundation | | Samuel L Pfaff |
| Sol Goldman Charitable Trust | | Samuel L Pfaff |

The funders had no role in study design, data collection and interpretation, or the decision to submit the work for publication.

### Author contributions

MJS, Designed the study, generated ES cell lines, conducted cell culture, ran calcium-imaging experiments, analyzed the data, and wrote the paper; CAH, NJM, Recorded electrophysiological activity; MTP, Generated an Hb9:GFP ES cell line and provided cell culture support; KLH, Generated the CAG:GCaMP3 mouse line; SPD, Designed and wrote analysis algorithms and analyzed the data; MH, Conducted immunohistochemistry; NDA, DB, WDG, Performed RNA sequencing; KS, MG, Provided mouse lines; SLP, Designed the study and wrote the paper

### Author ORCIDs

Samuel L Pfaff, http://orcid.org/0000-0002-2142-166X

### Ethics

Animal experimentation: All experiments were conducted in accordance with the Salk Institutional Animal Care and Use Committee Animal Protocols (IACUC #11-00020) and the Salk Institutional Review Board (IRB).

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
