## [Decision Letter]

Thank you for submitting your article "Circuit control mechanisms revealed through synthetic neural networks" for consideration by *eLife*. Your article has been favorably evaluated by a Senior Editor and three reviewers, one of whom, Ronald L Calabrese (Reviewer #1), is a member of our Board of Reviewing Editors and another one is Jeremy Dasen (Reviewer #2).

The reviewers have discussed the reviews with one another and the Reviewing Editor has drafted this decision to help you prepare a revised submission.

Title and Abstract: Please make the title more descriptive to conform to *eLife* standards. Synthetic is not sufficient: cultured and differentiated mouse embryonic stem cells should be mentioned. Abstract should echo the title and implement the major points of the required revision.

Summary:

This is a singular study in its approach. It uses embryonic stem cells that have cell type specific reporter genes and FACS to generate cultures of spinal interneurons and motor neurons of prescribed composition and then uses genetically expressed Ca indicators and electrophysiology to monitor activity and synaptic properties. It then creatively makes different circuitoids and dissects their circuit mechanisms for generating spontaneous bursting by pharmacological and genetic cell killing methods. The paper is brilliant in its concept and a great pleasure to read and think about. Its main conclusion is that strong excitatory connections lead to robust bursting and that when such strong interconnections are present then local inhibition promotes synchrony and decreasing the E/I ratio speeds bursting (decreases period). Conversely weak excitatory connections lead to less robust bursting and when such weak interconnections dominate the circuitoids then local inhibition disrupts synchrony and decreasing the E/I ratio leads to fractionation of the network into co-active subcircuits. The thoroughness and systematics of the approach are impressive and the results are nicely quantified and clearly presented in the figures. This paper should be read by all who are interested in brain circuits.

Essential revisions:

There are some concerns that must be addressed, but overall this appears to be a strong study of general significance.

1) The authors completely ignore structured inhibition in their Discussion, indeed throughout the paper. This is perhaps understandable because in the circuitoids inhibition is unstructured – random. You must redirect the entire paper's conceptual framework to embrace the concept that basic antagonisms that are so characteristic of CPGs require a structured pattern inhibitory connectivity. The authors produce something akin to Pearson's flexor burst generator or the PreBotz complex of the respiratory CPG, i.e., a neuronal oscillator, rather than a CPG which has both rhythm and patterning components. We would also argue that some antagonisms are probably hard wired and not that plastic to descending selection. Some of course are easily selected in/out but still are dependent on a structured pattern of inhibitory connectivity. The network fractionation that the authors observe when adding V1 interneurons to MN cultures is not equivalent to building basic antagonisms such as flexor extensor antagonisms or, in axial locomotion, side-to-side antagonisms. Both in Introduction and in Discussion this limitation of the approach should be forthrightly dealt with. The authors have not built a CPG in a dish and they can recognize this limitation without losing the inherent interest of the paper.

2) The actual synaptic connectivity and electrical activity of the 'circuitoids' are not demonstrated. We are concerned that there are real synapses present and that the neurons are spiking and transmitting signals down axons to release sites. At the beginning of Results the authors state "To investigate whether matured neurospheres likewise produce spontaneous bursts we monitored network output using both an extracellular suction recording electrode and calcium imaging." We would like to see some of these extracellular suction electrode recordings, particularly in conjunction with Ca-imaging, so that we can judge what the electrical activity of the circuitoids is like. The whole cell patch recordings of Figure 7 give some confidence that there are synapses – concomitant suction electrode recordings of the bursts would certainly increase that confidence – but some anatomical corroboration seems necessary. This could be done at the LM level with, e.g., with immunocytochemistry to show vGluT2 boutons in apposition to glutamatergic receptors, or at minimum in apposition to post-synaptic neurons.

3) Figure legends are overwhelming. Can some of this information be artfully incorporated into the text?

4) Quantification: There is no clear indication of the numbers of cultures / experiments done in any section. Beginning in Figure 6 (subsection “Inhibition decouples motor neuron networks into separate units”, second paragraph), the authors switch to expressing their cultures as ratios of different neuronal types. This is likely a more appropriate way to tell us what's going on, and should be done throughout. It is more specific and quantitative.

5) The authors mention performing RNAseq on the ES lines they purified and comparing to in vivo gene profiles (subsection “de novo generation of spinal cord neurons”, last paragraph). From the figure it appears this comparison was only performed for MNs and V2a INs (not V1 and V3 as implied by the text). Is this correct?

6) Much of the data in the paper relies on calcium imaging to measure the firing properties of circuitoids. One has to read the figure legends to understand how these assays were performed. We think the authors should include in the main text a description of how the GCAMP reporter lines were made (using a constitutive CAG:GCAMP3 line).

7) The role of electrical coupling in the circuitoids is not discussed when we know that such coupling exists in development.

8) In the Discussion, the caveat that the cells differentiated in culture may not have all the same features as the same cells in vivo should be mentioned.

[Editors' note: further revisions were requested prior to acceptance, as described below.]

Thank you for resubmitting your work entitled "Speed and pattern control mechanisms characterized in rhythmically-active circuits created from spinal neurons produced from genetically-tagged embryonic stem cells" for further consideration at *eLife*. Your revised article has been favorably evaluated by a Senior editor, a Reviewing editor, and two reviewers.

The manuscript has been improved but there are some remaining issues that need to be addressed before acceptance, as outlined below:

The authors have made a strong revision and the expert reviewers' previous concerns are largely met. The major remaining issue, which should be addressed is that of patterning. Because in CPG networks patterning and rhythm generation are often done by different circuit elements and are often conceptualized differently it is best not to confuse the issue by referring to the segmentation of activity in different cells in a circuitoid as patterning because coordinated stable phase relations are not established. The authors should not refer to pattern in their title and should precisely define what they mean by pattern: segmentation of circuitoid activity without fixed coordination. Reviewer #3's comments should all be addressed in the further revision, which will not require his re-review.

*Reviewer #3:*

1) Throughout the manuscript, from the title on, the authors refer not only to speed/rhythm but also to pattern. In the locomotor field, to which the authors are directing their study, the word "pattern" refers to the coordination of muscle groups or motoneuron pools. There is no evidence presented in this manuscript that supports such coordination. The authors do show data related to "sub-networks" in which they show that when one group of MNs burst, another group may not. But this is far from any pattern per se, and could have many explanations (threshold effect, synaptic failure in a sparsely innervated system, as 2 examples). My view is that the word and concept of "pattern" should be eliminated from this manuscript, and especially from title.

2) The authors are very focused throughout the manuscript on the "excitatory to inhibitory ratio" (or E/I ratio). This disregards the potential importance of neural types, i.e. any characteristic other than neurotransmitter. If the effects were simply E/I ratio, then adding more of any type of inhibitory cell (e.g. V2b, cortical inhibitory interneurons) would have the same effect as adding V1's. Do they really think that the identity of the cells has nothing to do with the findings?

3) The authors use the term "behaviorally relevant" – e.g. in the first paragraph of the subsection “Circuitoids produce coordinated bursts of activity” – why? Is a burst every 25s behaviorally relevant? This comes up again in Figure 2, which illustrates a burst every ~7 seconds – this is not "similar in frequency to fictive locomotor preparations" (see last paragraph of the aforementioned subsection, and subsection “Circuitoids and Central Pattern Generators”, third paragraph), which would be about 4 times faster (based on a perusal of Kiehn's work). There is no need to overstate the results, which are interesting enough on their own.

4) The term "synaptic strength" is not used properly. This term is not non-specific, and should refer to the size (not frequency, subsection “Inhibitory V1 interneurons control motor neuron burst rate via V3 interneurons”, last paragraph) of a PSC in response to stimulation of a presynaptic neuron. This has not been systematically quantified/analyzed. Usage includes: Discussion, first paragraph, where it is hyphenated; subsection “Circuitoids and Central Pattern Generators”, first paragraph (which also refers to EPSPs instead of EPSCs); Figure 8; and subsection “A cellular E/I model to produce flexible motor behaviors”, first paragraph. Also, is it that the connections are weak, or that the cell types are not able to support rhythmicity autonomously? For example, the MNs don't have SK currents (Miles et al., 2006), which could be needed for the response.

5) On a related note, what does "without exception displayed features similar to their in vivo counterparts" mean? This is not sufficient. The cells have some molecular markers in common with their in vivo counterparts, but who knows about their electrical properties that form the basis of the findings here?

---

## [Author Response]

*Title and Abstract: Please make the title more descriptive to conform to eLife standards. Synthetic is not sufficient: cultured and differentiated mouse embryonic stem cells should be mentioned. Abstract should echo the title and implement the major points of the required revision.*

Revised Title: “Speed and pattern control mechanisms characterized in rhythmically-active circuits created from spinal neurons produced from genetically-tagged embryonic stem cells”.

*[…] Essential revisions:*

*There are some concerns that must be addressed, but overall this appears to be a strong study of general significance.*

*1) The authors completely ignore structured inhibition in their Discussion, indeed throughout the paper. This is perhaps understandable because in the circuitoids inhibition is unstructured – random. You must redirect the entire paper's conceptual framework to embrace the concept that basic antagonisms that are so characteristic of CPGs require a structured pattern inhibitory connectivity. The authors produce something akin to Pearson's flexor burst generator or the PreBotz complex of the respiratory CPG, i.e., a neuronal oscillator, rather than a CPG which has both rhythm and patterning components. We would also argue that some antagonisms are probably hard wired and not that plastic to descending selection. Some of course are easily selected in/out but still are dependent on a structured pattern of inhibitory connectivity. The network fractionation that the authors observe when adding V1 interneurons to MN cultures is not equivalent to building basic antagonisms such as flexor extensor antagonisms or, in axial locomotion, side-to-side antagonisms. Both in Introduction and in Discussion this limitation of the approach should be forthrightly dealt with. The authors have not built a CPG in a dish and they can recognize this limitation without losing the inherent interest of the paper.*

We have addressed this point by adding an additional statement in the Discussion (subsection “Circuitoids and Central Pattern Generators”, second paragraph). The reviewer is correct, that we have not constructed a CPG that has control features for left-right and flexor-extensor coordination based on reciprocal inhibition (i.e. structured inhibition). To be clear, we have generated a rhythmic oscillatory network (emphasized by stating in the new title) that has operational similarities to a half center CPG in the lumbar spinal cord or a PreBotz CPG for respiration in the brainstem as recognized by the reviewer.

*2) The actual synaptic connectivity and electrical activity of the 'circuitoids' are not demonstrated. We are concerned that there are real synapses present and that the neurons are spiking and transmitting signals down axons to release sites. At the beginning of Results the authors state "To investigate whether matured neurospheres likewise produce spontaneous bursts we monitored network output using both an extracellular suction recording electrode and calcium imaging." We would like to see some of these extracellular suction electrode recordings, particularly in conjunction with Ca-imaging, so that we can judge what the electrical activity of the circuitoids is like. The whole cell patch recordings of Figure 7 give some confidence that there are synapses – concomitant suction electrode recordings of the bursts would certainly increase that confidence – but some anatomical corroboration seems necessary. This could be done at the LM level with, e.g., with immunocytochemistry to show vGluT2 boutons in apposition to glutamatergic receptors, or at minimum in apposition to post-synaptic neurons.*

Neurons within circuitoids fire as a coordinated unit, suggesting that they are interconnected. —As requested, we have added data showing immunostaining with pre-synaptic marker vGlut2 and post-synaptic marker PSD95 to demonstrate abundant synaptic structures form within circuitoids (see new Figure 1—figure supplement 3). The evidence for synaptic signaling within circuitoids is now demonstrated in three ways: (1) glutamatergic antagonists prevent bursting (Figure 2; Figure 4), (2) functional recordings of EPSPs in Figure 7, and (3) anatomical evidence of synaptic structures (new Figure 1—figure supplement 3).

As requested, we have also added more examples of electrical recordings in conjunction with Ca-imaging (Figure 1, and new Figure 1—figure supplement 3).

*3) Figure legends are overwhelming. Can some of this information be artfully incorporated into the text?*

Figure legends have been edited for clarity and brevity.

*4) Quantification: There is no clear indication of the numbers of cultures / experiments done in any section. Beginning in Figure 6 (subsection “Inhibition decouples motor neuron networks into separate units”, second paragraph), the authors switch to expressing their cultures as ratios of different neuronal types. This is likely a more appropriate way to tell us what's going on, and should be done throughout. It is more specific and quantitative.*

Sample and culture numbers are indicated in the figure legends together with p values. In general, sample sizes and recording periods are very large, particularly in comparison to typical in vivo experiments. The results with purified and inter-mixed neuron types begin in Figure 5. Unfortunately, it is not possible to list cell type ratios before this point in the paper.

*5) The authors mention performing RNAseq on the ES lines they purified and comparing to* in vivo *gene profiles (subsection “*de novo *generation of spinal cord neurons”, last paragraph). From the figure it appears this comparison was only performed for MNs and V2a INs (not V1 and V3 as implied by the text). Is this correct?*

The reviewer understands this correctly. To avoid confusion, we have simplified Figure 1—figure supplement 2 to include only the data from RNAseq of motor neurons and V2a interneurons. The evidence that V1 and V3 neurons are specified correctly is two fold: (1) they express the canonical marker for these neurons (En1 and Sim1, respectively) and (2) they display the correct pharmacological properties (glycinergic and glutamatergic, respectively). The text has been revised accordingly (subsection “De novo generation of spinal cord neurons”, last paragraph).

*6) Much of the data in the paper relies on calcium imaging to measure the firing properties of circuitoids. One has to read the figure legends to understand how these assays were performed. We think the authors should include in the main text a description of how the GCAMP reporter lines were made (using a constitutive CAG:GCAMP3 line).*

As recommended, we have explained the electrical and optical recording procedures in the main text (subsection “Circuitoids produce coordinated bursts of activity”, first paragraph) and in more detail in the Methods.

*7) The role of electrical coupling in the circuitoids is not discussed when we know that such coupling exists in development.*

We state that gap junctions appear to be insufficient to drive the sustained activity we see in our cultures (subsection “Excitatory interneuron networks produce rhythmic activity”, second paragraph), based on the finding that antagonists of glutamatergic synaptic activity (CNQX) profoundly inhibit activity. To further emphasize this point, we added a statement about gap junctions in the Discussion (subsection “Circuitoids and Central Pattern Generators”, first paragraph).

*8) In the Discussion, the caveat that the cells differentiated in culture may not have all the same features as the same cells in vivo should be mentioned.*

We added this caveat to the Discussion in the first paragraph of the subsection “Circuitoids and Central Pattern Generators”.

[Editors' note: further revisions were requested prior to acceptance, as described below.]

*The manuscript has been improved but there are some remaining issues that need to be addressed before acceptance, as outlined below:*

*The authors have made a strong revision and the expert reviewers' previous concerns are largely met. The major remaining issue, which should be addressed is that of patterning. Because in CPG networks patterning and rhythm generation are often done by different circuit elements and are often conceptualized differently it is best not to confuse the issue by referring to the segmentation of activity in different cells in a circuitoid as patterning because coordinated stable phase relations are not established. The authors should not refer to pattern in their title and should precisely define what they mean by pattern: segmentation of circuitoid activity without fixed coordination. Reviewer #3's comments should all be addressed in the further revision, which will not require his re-review.*

We replaced the term “pattern” with “segmentation” and included additional explanations at multiple places in the paper to clarify this point. We changed the title, Abstract, Introduction (last paragraph), Results (subsection “Inhibition decouples motor neuron networks into separate units”, second paragraph), and Discussion (subsection “Networks that produce rhythmic activity”, first paragraph and subsection “A cellular E/I model to produce flexible motor behaviors”, last paragraph).

*Reviewer #3:*

*1) Throughout the manuscript, from the title on, the authors refer not only to speed/rhythm but also to pattern. In the locomotor field, to which the authors are directing their study, the word "pattern" refers to the coordination of muscle groups or motoneuron pools. There is no evidence presented in this manuscript that supports such coordination. The authors do show data related to "sub-networks" in which they show that when one group of MNs burst, another group may not. But this is far from any pattern per se, and could have many explanations (threshold effect, synaptic failure in a sparsely innervated system, as 2 examples). My view is that the word and concept of "pattern" should be eliminated from this manuscript, and especially from title.*

See response to editor above. Where appropriate the word “pattern” has been replaced, defined, and/or explained throughout each section of the paper.

*2) The authors are very focused throughout the manuscript on the "excitatory to inhibitory ratio" (or E/I ratio). This disregards the potential importance of neural types, i.e. any characteristic other than neurotransmitter. If the effects were simply E/I ratio, then adding more of any type of inhibitory cell (e.g. V2b, cortical inhibitory interneurons) would have the same effect as adding V1's. Do they really think that the identity of the cells has nothing to do with the findings?*

Since we have not tested V2b interneurons or cortical inhibitory neurons, we cannot say. However, it has been argued that the V1 interneurons can be divided into ~50 subsets by the combinatorial expression of 19 identified transcription factors (Bikoff et al. 2016), suggesting that multiple inhibitory cell types share the characteristics defined in our assays. Please note that we have been careful to define V1 interneurons as inhibitory (I) and V3 interneurons and motor neurons as excitatory (E), and do not imply or speculate that other cell types are equivalent.

*3) The authors use the term "behaviorally relevant" – e.g. in the first paragraph of the subsection “Circuitoids produce coordinated bursts of activity” – why? Is a burst every 25s behaviorally relevant? This comes up again in Figure 2, which illustrates a burst every ~7 seconds – this is not "similar in frequency to fictive locomotor preparations" (see last paragraph of the aforementioned subsection, and subsection “Circuitoids and Central Pattern Generators”, third paragraph), which would be about 4 times faster (based on a perusal of Kiehn's work). There is no need to overstate the results, which are interesting enough on their own.*

In the first case the data refers to the spontaneous activity of circuitoids, which is consistent with the frequency of spontaneous ventral root depolarizations of a P2 mouse pup (Whelan et al. 2000: found spontaneous bursts to occur every 2.6 ± 1 (SD) episodes/min= 1 burst every 23s).

Additionally, referencing Figure 2, one has to remember that there are very few V1 inhibitory neurons present in these circuitoids (Figure 1), making them similar to En1:Cre;LSL:DTA spinal cords that have burst durations of about 10 seconds (Gosgnach et al. 2006). When we mixed V1 interneurons with V3 cells the burst rate dramatically increased. We recorded frequencies up to 0.45 Hz in V3/V1 networks in the presence of 100,000 V3 interneurons and 100,000 V1 interneurons (Figure 5 and data not shown). Therefore, the fact that circuitoids are (1) spontaneously active at a slow frequency comparable to the spinal cord, (2) become highly rhythmic in CPG-evoking drugs like the spinal cord, and (3) have burst frequencies around 0.5 Hz when inhibition is present which is not dissimilar to fictive spinal cord preps suggests circuitoid activity is relevant to locomotor behavior and not random.

*4) The term "synaptic strength" is not used properly. This term is not non-specific, and should refer to the size (not frequency, subsection “Inhibitory V1 interneurons control motor neuron burst rate via V3 interneurons”, last paragraph) of a PSC in response to stimulation of a presynaptic neuron. This has not been systematically quantified/analyzed. Usage includes: Discussion, first paragraph, where it is hyphenated; subsection “Circuitoids and Central Pattern Generators”, first paragraph (which also refers to EPSPs instead of EPSCs); Figure 8; and subsection “A cellular E/I model to produce flexible motor behaviors”, first paragraph. Also, is it that the connections are weak, or that the cell types are not able to support rhythmicity autonomously? For example, the MNs don't have SK currents (Miles et al., 2006), which could be needed for the response.*

We revised the description of the miniature post synaptic current data (Figure 7) as requested. We replaced the term “synaptic strength” with “network connection strength” to better describe the observation that mEPSC frequencies are higher among V3 interneurons and increase when V3 cells are added to motor neuron networks (subsection “Inhibitory V1 interneurons control motor neuron burst rate via V3 interneurons”, last paragraph, Discussion, first paragraph and subsection “Circuitoids and Central Pattern Generators”, Figure 8 legend). The data is quantified in Figure 7. As noted by the reviewer the reduced “network connection strength” we detect with motor neurons may be due to either weaker synaptic inputs or different channel properties intrinsic to motor neurons. We do not speculate on the mechanism. Thank you for helping us to correct this point.

*5) On a related note, what does "without exception displayed features similar to their in vivo counterparts" mean? This is not sufficient. The cells have some molecular markers in common with their in vivo counterparts, but who knows about their electrical properties that form the basis of the findings here?*

The reviewer is referring to the genetic profiling, cardinal marker expression, and neurotransmitter expression of the ES-derived neurons: "Although the ES cell-derived neuron subtypes used in this analysis without exception displayed features similar to their in vivo counterparts, we cannot exclude that there are differences between the in vitro and in vivo cells". In other words, we found no differences between the in vivo and in vitro cell types using these assays, but recognize that the ES-derived cells may not be identical in every regard to their in vivo counterparts.